# ON MEASURING INFLUENCE IN AVOIDING UNDESIRED FUTURE

**Lue Tao, Tian-Zuo Wang, Yuan Jiang, Zhi-Hua Zhou**
National Key Laboratory for Novel Software Technology, Nanjing University, China
School of Artifcial Intelligence, Nanjing University, China
`{taol,wangtz,jiangy,zhouzh}@lamda.nju.edu.cn`

## ABSTRACT

When a predictive model anticipates an undesired future event, a question arises: What can we do to avoid it? Resolving this forward-looking challenge requires determining the variables that positively influence the future, moving beyond the statistical *association* typically exploited for prediction. In this paper, we introduce a novel measure for evaluating the *influence* of actionable variables in successfully avoiding the undesired future. We quantify influence as the degree to which the success probability can be increased by altering variables under the principle of maximum expected utility. Our analysis demonstrates a counterintuitive insight: while related to *causality*, influential variables may not necessarily be those with strong intrinsic causal effects on the target event. In fact, it can be highly beneficial to alter a weak causal factor, or even a variable that is not an intrinsic factor at all. We provide a practical implementation for estimating the proposed measure and validate its utility through experiments on synthetic and real-world tasks.

## 1 INTRODUCTION

When an intelligent machine receives a warning from a powerful predictive model that an undesired event is expected to occur, an important question arises: What can be done to avoid this future? This is known as the *avoiding undesired future* (AUF) problem (Zhou, 2022b), sparking a shift from passively predicting eventualities to proactively shaping the future.

Addressing the AUF problem requires identifying which actions should be taken to realize a more desired future. While machine learning (ML) has achieved remarkable success in prediction (Jumper et al., 2021; Achiam et al., 2023; Price et al., 2025) by exploiting statistically associated variables to predict the target variable, the *association* alone is insufficient for guiding decision actions. *Causality* (Pearl, 2009; Peters et al., 2017) is more informative, but should not be regarded as necessary for

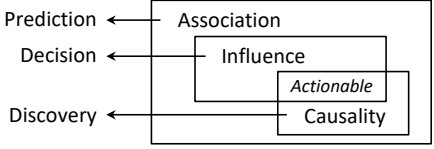

Figure 1: Relationship among association, influence, and causality (Zhou, 2022b).

decision-making (Zhou, 2022b) for reasons that (i) making decisions do not need a thorough characterization of causality, i.e., one can make good decisions without a full or faithful causal understanding; (ii) the real environment is often open and dynamic (Zhou, 2022a), so a cause of the target variable in historical data does not imply that altering it will influence the target in the future; (iii) discovering true causal relations from data is inherently difficult (Chickering, 1996), and even when identified, they are useless if unactionable. In light of this, Zhou (2022b; 2023) proposed the concept of *influence*, an intermediate that is stronger than association but less demanding than causality (Figure 1), forming the foundation for decision-making in AUF. Subsequently, a line of *rehearsal learning* methods have been developed to model influence among variables for decision-making, and have proven effective across a variety of AUF scenarios (Qin et al., 2023; 2025; Du et al., 2024; 2025a;b).

Despite these advances, a fundamental question remains open: *how to quantify influence in AUF?* Namely, *how to evaluate the influence of an actionable variable on the future target?* Conventional measures of causality (Rosenbaum & Rubin, 1983; Holland, 1988; Northcott, 2008) cannot serve this purpose, because they seek to evaluate the isolated effect of intervention within a static environment,

whereas AUF concerns a changeable world potentially reshaped by decisions. Accordingly, a causal strength evaluated on historical data may no longer indicate the actual strength of influence on the future target. Therefore, a principled quantification of influence is required for the AUF problem.

In this paper, we introduce a novel measure for evaluating the influence of actionable variables in successfully avoiding the undesired future. We begin by outlining several intuitive considerations that a measure of influence in AUF scenarios should incorporate. Then, we formulate a novel quantity, termed *influence power*, defined as the degree to which the probability of success can be increased through alteration based on the principle of maximum expected utility. This quantity captures the probabilistic dynamics of the decision process by accounting for the interplay between subsequent alterations and unfolding observations. Consequently, it provides a holistic valuation of the influence of alteration, encapsulating both its explicit and implicit impacts on the future target.

Next, we leverage the proposed quantity to elucidate the connection between influential variables and those with intrinsic causal relations to the target variable. Through a systematic analysis, we uncover a subtle yet important distinction: *influential variables are not simply a subset of causal ancestors, and vice versa.* Specifically, while related to causal effects, variables with positive influence power are not necessarily those with intrinsically strong causal effects. In fact, due to the dynamics of the decision process, it can be highly beneficial to alter a causal ancestor with negligible effects, or even a variable that is not an intrinsic ancestor at all. Another important observation is that not all actionable variables can be safely altered; there exist variables on which any alteration is counterproductive. These insights crystallize the Shakespearean quandary for an intelligent machine facing an eventuality: *To do, or not to do, that is the question.* This work rests on influence power, a principled quantity for measuring influence in AUF, thereby offering a rigorous way to answer this question.

Finally, we provide a Monte-Carlo-based implementation for estimating influence power using observational data. We empirically validate the utility of our measure in the AUF problem through experiments on synthetic and real-world tasks.

## 2 PRELIMINARY

**Notation.** We represent random variables with capital letters ($V$), and their realized values with lowercase letters ($v$). We use bold capital letters ($\mathbf{V}$) to denote a set of random variables, and bold lowercase letters ($\mathbf{v}$) to denote their realized values. Let $G = (\mathbf{V}, \mathbf{E})$ denote a directed graph with nodes $\mathbf{V}$ and edges $\mathbf{E}$. In a causal graph $G$, a variable $X$ is a causal ancestor of $Y$, written as $X \in \mathrm{Anc}(Y)$, if there is a directed path from $X$ to $Y$ in $G$. When $X$ is binary, its causal strength can be quantified by the *average causal effect* (ACE) (Holland, 1988; Pearl, 2009), defined as $\tau(X, Y) := \mathbb{E}(Y|do(X = 1) - \mathbb{E}(Y|do(X = 0)))$, where $\mathbb{E}(Y|do(X = x))$ denotes the expectation of $Y$ when $X$ is set to the value $x$. We say that a causal ancestor $X$ of $Y$ is weak if the average causal effect of $X$ on $Y$ is zero. Let $\Delta_X$ denote the feasible domain of alteration for a variable $X$. If $\Delta_X \neq \emptyset$, we call $X$ an actionable variable; otherwise, $X$ is unactionable.

**Structural equation models.** We employ the language of the *structural equation model* (SEM) (Pearl, 2009) and leverage it to describe how *nature* assigns values to variables of interest, i.e., the physical mechanisms governing the generation process of random variables. An SEM is a tuple $\mathcal{M} = \langle \mathbf{V}, \mathbf{N}, F, P(\mathbf{N}) \rangle$, where $\mathbf{V} = \{V_1, \ldots, V_d\}$ is a set of endogenous variables, $\mathbf{N} = \{N_1, \ldots, N_d\}$ is a set of background noises distributed according to $P(\mathbf{N})$, and $F$ is a set of deterministic functions $f_i$ for each $V_i \in \mathbf{V}$ such that $V_i := f_i(\mathrm{PA}_i, N_i)$ with $\mathrm{PA}_i \subseteq \mathbf{V}$. Throughout this paper, we posit that the natural generation process is governed by an underlying SEM $\mathcal{M}$, though it may remain unknown to the decision-maker due to its unobserved nature (Geffner et al., 2022). If $V_i$ has a directed path to $V_j$ in the graph induced by $\mathcal{M}$, we say that $V_i$ is an intrinsic ancestor of $V_j$ in $\mathcal{M}$. For a variable $V_i$, when it is actionable, we use the notation $V_i \overset{a}{=} v_i$ to indicate that $V_i$ can be *altered* to $v_i \in \Delta_{V_i}$; in contrast, the notation $do(V_i = v_i)$ is applicable even when $V_i$ is not actionable. This operation essentially replaces the structural function of $V_i$ in $\mathcal{M}$ with the constant assignment $V_i := v_i$. The distribution of $V_j$ given that $V_i$ is altered to $v_i$ is then denoted as $P(V_j | V_i \overset{a}{=} v_i)$.

**Problem statement.** We consider a setting where data are drawn from a distribution induced by an underlying SEM, which describes the natural generation process of a sequence of variables $(V_1, \ldots, V_{d+1})$. The final variable $V_{d+1}$ in the sequence represents the target variable $Y$, with the set of desired values specified as $\mathcal{S}$. We assume for simplicity that all variables are discrete, and the

variable sequence is pre-specified and consistent with the underlying variable ordering (i.e., variables are topologically ordered with respect to the SEM). The goal of the AUF problem is to maximize the probability of $Y$ falling into $\mathcal{S}$ through feasible alterations to the variables $V_1, \ldots, V_d$. In general, decision-making occurs after certain variables have already been realized. We denote this subset of realized variables as $\mathbf{X} \subseteq \{V_1, \ldots, V_d\}$. Let $t = \max\{s \mid V_s \in \mathbf{X}\}$ be the index of the latest realized variable, and denote the remaining actionable variables as $\mathbf{Z} = \{V_i \mid t < i \leq d, \Delta_{V_i} \neq \emptyset\}$.[1] Hence, given a realization of $\mathbf{X}$, the AUF problem is addressed by altering variables in $\mathbf{Z}$. Notably, when determining the alteration for a specific variable $V_i$, we do not require all variables preceding $V_i$ to be realized. This flexibility allows for certain variables to remain unobserved during the decision process, which is more practical than requiring full observability.

## 3 INFLUENCE POWER

### 3.1 MOTIVATION

We review the existing strategies for addressing the AUF problem, describe their limitations, and motivate the considerations that a measure of influence in AUF should incorporate. A primary AUF strategy in the literature is to find a feasible alteration that directly maximizes the probability that $Y$ lies in the desired domain $\mathcal{S}$ (Qin et al., 2023). This straightforward strategy is expressed as:

$$(Z^*, z^*) = \arg\max_{Z \in \mathbf{Z}, z \in \Delta_Z} P(Y \in \mathcal{S} | \mathbf{X} = \mathbf{x}, Z \overset{a}{=} z), \tag{1}$$

where $\mathbf{x}$ is the observed value of $\mathbf{X}$, $\mathbf{Z}$ is the set of actionable variables following $\mathbf{X}$, and $P(Y \in \mathcal{S} | \mathbf{X} = \mathbf{x}, Z \overset{a}{=} z)$ is the AUF probability (i.e., the probability of $Y \in \mathcal{S}$) after observing $\mathbf{X}$ as $\mathbf{x}$ and altering $Z$ to $z$. While intuitive and often effective, Equation (1) evaluates only the explicit impact of altering one variable at a time, implicitly presuming a "static" future in which subsequent variables unfold naturally without further alteration. As a result, it fails to capture the impact stemming from the interplay between altering variables, overlooking situations where no single alteration suffices to achieve the desired target. A simple example illustrates this issue. Consider two binary variables, $Z_1$ and $Z_2$, both of which naturally take the value 0 with near certainty, and let $Y := Z_1 \wedge Z_2$. Altering either variable in isolation is ineffective; only by altering both variables jointly can we ensure $Y = 1$. Thus, when assessing the impact of altering a variable in AUF scenarios, one must consider not only the feasible domain of that alteration but also the actionability of other variables.

In light of this insight, the next plausible step is to jointly alter all actionable variables. This approach is considered in prior work (Qin et al., 2025; Du et al., 2025a) and can be written as:

$$\mathbf{z}^* = \arg\max_{\mathbf{z} \in \Delta_{\mathbf{z}}} P(Y \in \mathcal{S} | \mathbf{X} = \mathbf{x}, \mathbf{Z} \overset{a}{=} \mathbf{z}), \tag{2}$$

where $\Delta_{\mathbf{z}}$ denotes the Cartesian product of the feasible domains of alteration for variables in $\mathbf{Z}$, and $\mathbf{Z} \overset{a}{=} \mathbf{z}$ denotes the joint alteration of variables in $\mathbf{Z}$ to the values in $\mathbf{z}$. This strategy also overlooks an important consideration: it is often unnecessary to alter all variables. For example, while both light and water are crucial for crop growth, adding artificial light when sunlight is naturally abundant yields negligible benefits. Thus, when assessing the impact of altering a variable in AUF scenarios, one should account for its naturality, i.e., whether it naturally lies in a favorable state. Moreover, as we will demonstrate later, altering certain variables may be not only unnecessary but counterproductive, no matter how they are altered. These observations underscore the need for a principled measure to determining whether an actionable variable warrants alteration.

### 3.2 FORMULATION

In this subsection, we present a new quantity that assesses whether an actionable variable is worth altering to influence the future. To holistically account for the actionability and naturality of variables, as well as the desirability of the target variable, we need a principled way to rehearse future possibilities after an alteration. While the Bellman equation (Bellman, 1957) provides a conceptual foundation for this purpose, its standard formulation is not applicable to our context, as it is typically built upon a pre-specified separation between state and control variables. In the AUF problem, however, every variable $V_i$ in the sequence $(V_1, \ldots, V_d)$ has a dual role: it can be proactively manipulated via alteration or passively observed as it unfolds naturally.

---

[1] As the past cannot be changed, variables preceding $V_t$ are considered not actionable.

To this end, we ground our proposal in the principle of maximum expected utility (Russell & Norvig, 2020), and recursively define the *maximum expected probability* (MEP) of achieving the desired future after an alteration or observation. Specifically, for $0 < k < d$, the MEP after *altering* $V_k$ to $v_k$, denoted as $\mathcal{P}(Y \in \mathcal{S}|V_k \stackrel{a}{=} v_k, \ldots)$, is given by:

$$\mathcal{P}(Y \in \mathcal{S}|V_k \stackrel{a}{=} v_k, \ldots) = \max \Big\{ \max_{v_{k+1} \in \Delta_{V_{k+1}}} \mathcal{P}(Y \in \mathcal{S}|V_{k+1} \stackrel{a}{=} v_{k+1}, V_k \stackrel{a}{=} v_k, \ldots), \tag{3}$$
$$\mathbb{E}_{v_{k+1} \sim P(V_{k+1}|V_k \stackrel{a}{=} v_k, \ldots)} \mathcal{P}(Y \in \mathcal{S}|V_{k+1} = v_{k+1}, V_k \stackrel{a}{=} v_k, \ldots) \Big\},$$

where "$\ldots$" abbreviates an arbitrary sequence of alterations and observations preceding $V_k$. For the terminal step $k = d$, the recursion terminates and no further decomposition is required; the MEP simply reduces to the ordinary probability, i.e., $\mathcal{P}(Y \in \mathcal{S}|V_k \stackrel{a}{=} v_k, \ldots) = P(Y \in \mathcal{S}|V_k \stackrel{a}{=} v_k, \ldots)$.

Similarly, for $0 < j < d$, the MEP after *observing* $V_j$ as $v_j$ is given by:

$$\mathcal{P}(Y \in \mathcal{S}|V_j = v_j, \ldots) = \max \Big\{ \max_{v_{j+1} \in \Delta_{V_{j+1}}} \mathcal{P}(Y \in \mathcal{S}|V_{j+1} \stackrel{a}{=} v_{j+1}, V_j = v_j, \ldots), \tag{4}$$
$$\mathbb{E}_{v_{j+1} \sim P(V_{j+1}|V_j = v_j, \ldots)} \mathcal{P}(Y \in \mathcal{S}|V_{j+1} = v_{j+1}, V_j = v_j, \ldots) \Big\},$$

where "$\ldots$" abbreviates an arbitrary sequence of alterations and observations preceding $V_j$. Likewise, as the sequence concludes at $j = d$, the MEP after observing $V_j$ as $v_j$ directly equals the ordinary probability, i.e., $\mathcal{P}(Y \in \mathcal{S}|V_j = v_j, \ldots) = P(Y \in \mathcal{S}|V_j = v_j, \ldots)$.

Given the recursive definition of MEP, we present a quantity called the *influence power* (InP), which indicates the ability of an actionable variable to influence the target at a future time.

**Definition 1** (Influence Power). *The influence power of an actionable variable $V_i$ on $Y$ is defined as*

$$\dot{p}(V_i, Y) := \max_{v_i \in \Delta_{V_i}} \mathcal{P}(Y \in \mathcal{S}|V_i \stackrel{a}{=} v_i) - \mathbb{E}_{v_i \sim P(V_i)} \mathcal{P}(Y \in \mathcal{S}|V_i = v_i).$$

**Remark.** The influence power of $V_i$ on $Y$ quantifies the maximum gain in the MEP achievable by altering $V_i$, relative to the expected MEP when $V_i$ is observed naturally. Consequently, a positive value indicates that alteration is beneficial, while a zero or negative value suggests that it is unnecessary or even harmful. By definition, the influence power is bounded in $[-1, 1]$. Because Definition 1 recursively applies the principle of maximum expected utility, it can be interpreted as a variant of the Bellman equation. Notably, it depends only on probability terms and does not require a fully specified SEM. Furthermore, this definition readily extends to a conditional form: given the observation of $\mathbf{X} = \mathbf{x}$, the conditional influence power of $V_i \in \mathbf{Z}$ on $Y$, denoted as $\dot{p}(V_i, Y|\mathbf{X} = \mathbf{x})$, is given by $\max_{v_i \in \Delta_{V_i}} \mathcal{P}(Y \in \mathcal{S}|V_i \stackrel{a}{=} v_i, \mathbf{X} = \mathbf{x}) - \mathbb{E}_{v_i \sim P(V_i|\mathbf{X} = \mathbf{x})} \mathcal{P}(Y \in \mathcal{S}|V_i = v_i, \mathbf{X} = \mathbf{x})$.

**Relation to Equation (1).** We end this subsection by highlighting a connection between Definition 1 and Equation (1). Consider a scenario with three binary variables: $V_1$, $V_2$, and $Y$, where both $V_1$ and $V_2$ are actionable. Suppose an oracle informs us that the structural function $f$ defining the target variable $Y$ depends only on $V_1$ (and not on $V_2$), i.e., $Y := f(V_1)$. Based on this information, we deduce that the solution to Equation (1) is $V_1$ whenever the following condition holds:

$$\max_{v_1 \in \Delta_{V_1}} P(Y \in \mathcal{S}|V_1 \stackrel{a}{=} v_1) > P(Y \in \mathcal{S}). \tag{5}$$

On the other hand, the influence power of $V_1$ on $Y$ simplifies to:

$$\dot{p}(V_1, Y) = \max_{v_1 \in \Delta_{V_1}} P(Y \in \mathcal{S}|V_1 \stackrel{a}{=} v_1) - P(Y \in \mathcal{S}). \tag{6}$$

Combining Equations (5) and (6) we conclude that the solution to Equation (1) is $V_1$ if $\dot{p}(V_1, Y) > 0$. Thus, Equation (1) agrees with Definition 1 on whether $V_1$ should be altered.

Furthermore, let $\Delta_{V_1} = \{0, 1\}$ and $\mathcal{S} = \{1\}$. By applying the identity $2 \cdot \max(a, b) = a + b + |a - b|$, the condition $\dot{p}(V_1, Y) > 0$ reduces to:

$$|\tau(V_1, Y)| \equiv |\mathbb{E}(Y|V_1 \stackrel{a}{=} 1) - \mathbb{E}(Y|V_1 \stackrel{a}{=} 0)| > 2\mathbb{E}(Y) - \mathbb{E}(Y|V_1 \stackrel{a}{=} 0) - \mathbb{E}(Y|V_1 \stackrel{a}{=} 1), \tag{7}$$

where $|\tau(V_1, Y)|$ is the absolute value of average causal effect of $V_1$ on $Y$. This reveals that $\dot{p}(V_1, Y)$ is closely related to $\tau(V_1, Y)$, and the influence power seems to favor altering variables with strong causal effects. This view, however, is incomplete. In the following section, we demonstrate that the connection between influence power and average causal effect is, in fact, far more nuanced.

### 3.3 CONNECTION

In this subsection, we investigate the connection between influential variables and those with intrinsic causal relations to the target variable. Concretely, we analyze how variables with a non-zero influence power (i.e., $\dot{p}(X,Y) \neq 0$) relate to the intrinsic ancestors of the target in the underlying SEM (denoted by $\mathrm{Anc}(Y)$). We also examine the qualitative relationship between the influence power $\dot{p}(X,Y)$ and the average causal effect $\tau(X,Y)$, a commonly used measure of causal strength. The results are formally summarized in the following theorem.

**Theorem 1.** *Let $X$ and $Y$ be two endogenous variables in an SEM. The following statements hold:*

1. $X \in \mathrm{Anc}(Y) \;\not\!\!\Longrightarrow\; \dot{p}(X,Y) \neq 0$ *and* $\dot{p}(X,Y) \neq 0 \;\not\!\!\Longrightarrow\; X \in \mathrm{Anc}(Y)$;

2. $\tau(X,Y) \neq 0 \;\not\!\!\Longrightarrow\; \dot{p}(X,Y) \neq 0$ *and* $\dot{p}(X,Y) \neq 0 \;\not\!\!\Longrightarrow\; \tau(X,Y) \neq 0$;

3. $\tau(X,Y) \neq 0 \;\not\!\!\Longrightarrow\; \dot{p}(X,Y) \geq 0$ *and* $X \in \mathrm{Anc}(Y) \;\not\!\!\Longrightarrow\; \dot{p}(X,Y) \geq 0$.

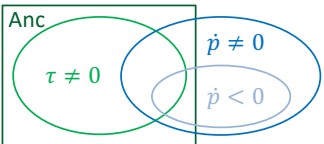

Theorem 1 reveals a nuanced relationship between intrinsic ancestors and non-zero influence power: neither implies the other. Specifically, a causal ancestor of the target can have zero influence power, and conversely, a variable with non-zero influence power is not necessarily an intrinsic ancestor in the underlying SEM. Similarly, a variable may have a non-zero average causal effect on the target while having zero influence power, and vice versa. Furthermore, neither having a non-zero average causal effect nor being a causal ancestor guarantees non-negative influence power.

Figure 2: Relationships among intrinsic ancestors, $\tau$, and $\dot{p}$.

These relationships are visualized in the Venn diagram in Figure 2. In what follows, we demonstrate several key statements from Theorem 1 with concrete examples, and we defer the detailed proof to Appendix B. For clarity, these demonstrations focus on binary variables with the desired domain $\mathcal{S} = \{1\}$, though these restrictions are not required in general.

**A causal ancestor can have zero influence power.** Altering a causal ancestor with a strong average causal effect may yield no benefit for the target.

**Example 1.** *Consider the following structural equations and the induced graph:*
$$X := N_X,$$
$$Z := X \cdot N_Z + (1 - X) \cdot (1 - N_Z),$$
$$Y := Z \cdot N_Y + (1 - Z) \cdot (1 - N_Y),$$

$$X \rightarrow Z \rightarrow Y$$

*where $N_X, N_Z, N_Y \overset{iid}{\sim} \mathrm{Bern}(0.9)$. Let $X$ and $Z$ be actionable variables, let $\Delta_X = \{0,1\}$ and $\Delta_Z = \{0,1\}$ be their feasible domains of alteration, and let the desired domain for $Y$ be $\mathcal{S} = \{1\}$.*

In this example, while $X$ is an ancestor of $Y$ in the SEM, its influence power on $Y$ is zero: $\dot{p}(X,Y) = \max_{x \in \Delta_X} \mathcal{P}(Y = 1 | X \overset{a}{=} x) - \mathbb{E}_{x \sim P(X)} \mathcal{P}(Y = 1 | X = x) = \max_{x \in \Delta_X} \max_{z \in \Delta_Z} P(Y = 1 | Z \overset{a}{=} z, X \overset{a}{=} x) - \mathbb{E}_{x \sim P(X)} \max_{z \in \Delta_Z} P(Y = 1 | Z \overset{a}{=} z, X = x) = 0.9 - 0.9 = 0$. This indicates that altering $X$ yields no improvement in the probability of $Y = 1$; a rational machine will always maximize the probability of $Y = 1$ by setting $Z$ to 1, regardless of the value of $X$. In short, altering $X$ in Example 1 is useless because the actionability of $Z$ shields its effect. Thus, a causal ancestor does not necessarily have non-zero influence power. In addition, the average causal effect of $X$ on $Y$ in the SEM is non-zero: $\tau(X,Y) = P(Y = 1 | X \overset{a}{=} 1) - P(Y = 1 | X \overset{a}{=} 0) = 0.82 - 0.18 = 0.64$. This shows that a non-zero average causal effect does not guarantee non-zero influence power.

**A non-ancestral variable can have non-zero influence power.** Altering a variable that is not an intrinsic ancestor of the target in the underlying SEM may still benefit the target.

**Example 2.** *Consider the following structural equations and the induced graph:*
$$U := N_U,$$
$$W := N_W,$$
$$X := U \cdot W \cdot (1 - N_X),$$
$$Z := N_Z,$$
$$Y := Z \cdot (1 - U) + (1 - Z) \cdot N_Y,$$

where $N_U \sim \text{Bern}(0.5)$, $N_W, N_X, N_Z \stackrel{iid}{\sim} \text{Bern}(0.1)$, and $N_Y \sim \text{Bern}(0.4)$. Let $W$, $X$, and $Z$ be actionable variables with $\Delta_W = \Delta_X = \Delta_Z = \{0, 1\}$, and let the desired domain be $\mathcal{S} = \{1\}$.

In this example, $W$ is not an ancestor of $Y$ in the SEM. Notably, the influence power of $W$ on $Y$ is positive: $\dot{p}(W, Y) = 0.68 - 0.518 = 0.162$. This indicates that altering $W$ can significantly increase the MEP of $Y = 1$. Intuitively, this positive influence manifests because altering $W$ can help $X$ reveal information about $U$, which facilitates a more informed alteration of $Z$, ultimately benefiting $Y$. To provide concrete intuition, let us ground the variables from Example 2 in a medical scenario: let $U$, $W$, $X$, $Z$, and $Y$ denote an allergy gene, a skin test, the skin response, a drug injection, and patient recovery, respectively. Performing a skin test ($W$) has no therapeutic effect; thus, the average causal effect of $W$ on $Y$ is zero. Nevertheless, the skin test is crucial because it informs the doctor's decision to administer the drug ($Z$), which affects recovery ($Y$). For instance, if the skin test is positive (observing $X = 1$ after setting $W \stackrel{a}{=} 1$), the doctor can infer the presence of the allergy and decide not to administer the drug (by setting $Z \stackrel{a}{=} 0$), thereby maximizing the probability of recovery ($Y = 1$). This shows that while $W$ does not intrinsically cause $Y$, altering $W$ is instrumental in influencing $Y$. Influence power successfully captures this implicit benefit, indicating that even non-ancestral variables can be critical for AUF.

For completeness, we also examine the conditional influence power of $W$ on $Y$ given $U$. We find that $\dot{p}(W, Y|U = 1) = 0 = \dot{p}(W, Y|U = 0) = 0$. This implies that if the allergy gene ($U$) is observed, performing the skin test ($W$) will be unnecessary. In clinical practice, however, directly genotyping the allergy gene ($U$) for a new patient is often time-consuming or prohibitively expensive. Thus, the unconditional influence power offers valuable guidance for addressing the AUF problem in practice.

**A weak ancestor can have positive influence power.** Altering a causal ancestor with a negligible average causal effect on the target may still benefit the target.

**Example 3.** *Consider the following structural equations and the induced graph:*

$$X := N_X,$$
$$Z := (1 - X) \cdot N_Z,$$
$$Y := X \cdot Z \cdot N_Y,$$

where $N_X, N_Z, N_Y \stackrel{iid}{\sim} \text{Bern}(0.5)$. Let $X$ and $Z$ be actionable variables with $\Delta_X = \{0, 1\}$ and $\Delta_Z = \{0, 1\}$, and let the desired domain for $Y$ be $\mathcal{S} = \{1\}$.

In this example, $X$ is an ancestor of $Y$, and the average causal effect of $X$ on $Y$ is zero: $\tau(X, Y) = 0$. Still, the influence power of $X$ on $Y$ is positive: $\dot{p}(X, Y) = 0.25$. Intuitively, this positive influence power manifests through the synergy between $X$ and $Z$. The benefit of $X$ on $Y$ is elicited when we account for the alteration of $Z$. This implicit impact is captured by $\dot{p}(X, Y)$ and missed by $\tau(X, Y)$.

**A strong ancestor can have negative influence power.** Altering a causal ancestor with a non-negligible average causal effect can be not only useless but also detrimental for the target.

**Example 4.** *Consider the following structural equations and the induced graph:*

$$U := N_U,$$
$$X := U \cdot N_X + (1 - U) \cdot (1 - N_X),$$
$$Z := X \cdot N_Z + (1 - X) \cdot (1 - N_Z),$$
$$Y := Z \cdot (1 - U) + (1 - Z) \cdot N_Y,$$

where $N_U \sim \text{Bern}(0.5)$, $N_X, N_Z \stackrel{iid}{\sim} \text{Bern}(0.9)$, and $N_Y \sim \text{Bern}(0.4)$. Let $X$ and $Z$ be actionable variables with $\Delta_X = \{0, 1\}$ and $\Delta_Z = \{0, 1\}$, and let the desired domain be $\mathcal{S} = \{1\}$.

In this example, $X$ is an ancestor of $Y$ in the SEM with a non-zero average causal effect: $\tau(X, Y) = 0.08$, but the influence power is negative: $\dot{p}(X, Y) = -0.15$. This indicates that the MEP after altering $X$ is lower than the expected MEP after observing $X$. Thus, any alteration on $X$ is counterproductive, regardless of the value to which $X$ is set. Intuitively, this negative influence manifests because observing $X$ reveals information about $U$, which is critical in determining the alteration of $Z$ during the computation of $\dot{p}(X, Y)$. Hence, although altering $X$ can produce a straightforward improvement in $Y$ (as indicated by the non-zero $\tau(X, Y)$), this benefit is outweighed by its negative impact on the subsequent alteration of $Z$, ultimately making the alteration of $X$ detrimental. This implicit impact is reflected in the value of influence power.

# 4 ESTIMATING INFLUENCE POWER

Influence power is a principled quantity for measuring the influence of actionable variables in AUF. In practice, exact computation can be intractable due to the need to exhaustively evaluate MEP terms; moreover, when the underlying SEM is unknown, the AUF probabilities needed for these terms are not directly available and must be estimated from observational data. To address these challenges, we provide a practical estimation method in this section.

## 4.1 MONTE-CARLO APPROXIMATION

Recursively enumerating MEP over all possible alterations can be computationally prohibitive when the number of actionable variables is large. To mitigate this, we cast MEP computation as a single-player non-deterministic game and approximate it using the Monte-Carlo tree search UCT (Upper Confidence Tree) introduced by Kocsis & Szepesvári (2006).

Specifically, we incrementally construct a *search tree* using Monte Carlo simulations. Each *node* in the tree represents a *state* defined by a sequence of alterations and observations made so far, associated with the next variable to be considered. Every iteration starts at the root node $N_0$ (associated with a pre-specified variable $V_i \in \mathbf{V}$), traverses to its children (associated with $V_{i+1}$), and continues until reaching a terminal state (associated with the target variable $Y$). Each *edge* in the tree represents a *choice* available at the node, i.e., either an alteration or an observation on the associated variable. The overall construction consists of four steps, iterated until the computational budget is exhausted: (1) *Selection*: start from the root node and recursively select an edge to child nodes according to the UCT policy until reaching a leaf node; (2) *Expansion*: if the leaf node corresponds to a non-terminal state, expand it by randomly adding one child node for each possible choice; (3) *Playout*: from the newly added node, execute a random sequence of choices until reaching a terminal state, and compute the AUF probability at that terminal state; (4) *Backpropagation*: propagate the computed AUF probability back up the tree, updating the statistics of each node along the path. During each iteration, the UCT criterion is used at a node $N$ to select the next edge to traverse:

$$c_N^* = \arg\max_{c \in \Delta_N^+} \left\{ \hat{p}_{N,c} + \alpha \cdot \sqrt{\frac{\ln t_N}{t_{N,c}}} \right\}, \tag{8}$$

where $\Delta_N^+ = \Delta_N \cup \emptyset$ is the set of choices at node $N$, consisting of feasible alterations on the variable associated with $N$ (i.e., $\Delta_N$) and the option to make an observation, (i.e., $\emptyset$). Here, $\hat{p}_{N,c}$ is the average AUF probability obtained after taking choice $c$ at node $N$, $\alpha$ is a parameter used to balance between exploration and exploitation (Auer et al., 2002), $t_N$ is the number of times node $N$ has been selected, and $t_{N,c}$ is the number of times choice $c$ has been selected at node $N$.

After constructing the search tree, we approximate the MEP terms in the influence power of $V_i$ on $Y$ by the average AUF probability for each choice at the root node $N_0$ of the search tree. Concretely, we have $\mathcal{P}(Y \in \mathcal{S} | V_i \overset{a}{=} c) \approx \hat{p}_{N_0,c}$ for each $c \in \Delta_{N_0}$, and $\mathbb{E}_{v_i \sim P(V_i)} \mathcal{P}(Y \in \mathcal{S} | V_i = v_i) \approx \hat{p}_{N_0,\emptyset}$. Hence, according to Definition 1, the influence power of $V_i$ on $Y$ is approximated as:

$$\dot{p}(V_i, Y) \approx \max_{c \in \Delta_{N_0}} \hat{p}_{N_0,c} - \hat{p}_{N_0,\emptyset}. \tag{9}$$

The quality of this approximation improves over time, as UCT converges to the best choice given sufficient iterations. Moreover, the described procedure is an *anytime* algorithm, capable of producing an approximate influence power at any point during its computation. We refer the reader to Browne et al. (2012) for additional details.

Finally, we emphasize an important practical phenomenon: Equation (9) can serve as a valuable indicator even with a limited number of Monte-Carlo simulations. This is because AUF does not always require a highly accurate estimate of influence power; in many cases, a coarse approximation is enough. Specifically, if the ground-truth influence power of a variable is non-positive ($\dot{p} \leq 0$), the approximation succeeds as long as it correctly indicates that no alteration of the variable is beneficial, which only requires $\hat{p}_{N_0,\emptyset} \geq \max_{c \in \Delta_{N_0}} \hat{p}_{N_0,c}$. Similarly, if the ground-truth influence power is positive ($\dot{p} > 0$), the approximation succeeds as long as it correctly identifies the optimal alteration $c^*$, i.e., it preserves the correct ordering of the MEP terms; it suffices that $\hat{p}_{N_0,c^*} \geq \max_{c \in \Delta_{N_0}} \hat{p}_{N_0,c}$. Our experiments provide empirical evidence for this phenomenon: with a relatively small number of simulations, the method can often yields the right "alter vs. observe" decision and the best alteration (see Section 5 for details). We believe this phenomenon merits consideration in future studies.

### 4.2 AUF Probability Estimation

While the Monte-Carlo procedure described above can effectively approximate the influence power, at terminal states during simulations, it still requires the AUF probability, whose ground-truth value is dictated by the underlying SEM. For situations where the structural equations are unknown, we present an estimator of the AUF probability from observational data.

Specifically, we express the joint probability of $(\mathbf{V}, Y)$ as:

$$P(\mathbf{V}, Y) = P(V_1, \ldots, V_d, Y) = P(Y|\mathbf{V}) \prod_{i=1}^{d} P(V_i|V_1, \ldots, V_{i-1}), \tag{10}$$

where the conditional probabilities $P(Y|\mathbf{V})$ and $P(V_i|V_1, \ldots, V_{i-1})$ can be estimated from observational data $\mathcal{D} = \{(\mathbf{v}^j, y^j)\}_{j=1}^{n}$ using standard ML models. Let $\mathbf{A}$ denote the variables in $\mathbf{V}$ that are altered. Then the joint probability of $(\mathbf{V}, Y)$ given the alteration of $\mathbf{A}$ is expressed as:

$$P(\mathbf{V}, Y|\hat{\mathbf{A}}) = P(Y|\mathbf{V}) \prod_{V_i \in \mathbf{A}} \delta(V_i) \prod_{V_i \in \mathbf{V} \setminus \mathbf{A}} P(V_i|V_1, \ldots, V_{i-1}), \tag{11}$$

where $\hat{\mathbf{A}}$ indicates that every variable $V_i \in \mathbf{A}$ is altered, and $\delta(\cdot)$ is the Dirac delta function. Let $\mathbf{O}$ denote the variables in $\mathbf{V}$ that are observed. Then the AUF probability given the alteration of $\mathbf{A}$ and the observation of $\mathbf{O}$ is expressed as:

$$
\begin{aligned}
P(Y \in \mathcal{S}|\hat{\mathbf{A}}, \mathbf{O}) &= \frac{P(Y \in \mathcal{S}, \mathbf{O}|\hat{\mathbf{A}})}{P(\mathbf{O}|\hat{\mathbf{A}})} = \frac{\sum_{\mathbf{V} \setminus \mathbf{O}} P(Y \in \mathcal{S}, \mathbf{V}|\hat{\mathbf{A}})}{\sum_{\mathbf{V} \setminus \mathbf{O}} P(\mathbf{V}|\hat{\mathbf{A}})} \\
&= \frac{\sum_{\mathbf{V} \setminus \mathbf{O}} P(Y \in \mathcal{S}|\mathbf{V}) \prod_{V_i \in \mathbf{A}} \delta(V_i) \prod_{V_i \in \mathbf{V} \setminus \mathbf{A}} P(V_i|V_1, \ldots, V_{i-1})}{\sum_{\mathbf{V} \setminus \mathbf{O}} \prod_{V_i \in \mathbf{A}} \delta(V_i) \prod_{V_i \in \mathbf{V} \setminus \mathbf{A}} P(V_i|V_1, \ldots, V_{i-1})},
\end{aligned}
\tag{12}
$$

which is a generic expression of the AUF probability under arbitrary alterations and observations. It can be estimated from observational data $\mathcal{D}$ and then plugged into the Monte-Carlo procedure described above to approximate the influence power. The following proposition shows the consistency of Equation (12) by applying the manipulation theorem from Spirtes et al. (2000).

**Proposition 1.** *Assume causal sufficiency, i.e., the joint distribution $P(\mathbf{V}, Y)$ is induced by an acyclic SEM $\mathcal{M}$ with independent noises. Also assume positivity, i.e., $P(V_i|\mathrm{PA}_i) > 0$ for all $1 \leq i \leq d$ on the support of $P$. Then, the expression in Equation (11) is consistent with the joint probability dictated by the SEM under the alteration of $\mathbf{A}$. Furthermore, the expression in Equation (12) is consistent with the AUF probability dictated by the SEM under the alteration of $\mathbf{A}$ and the observation of $\mathbf{O}$.*

**Remark.** While causal sufficiency is required for Proposition 1, it is not assumed for the remainder of the paper. This assumption is technically needed for reliably estimating the AUF probability in Equation (12) from observational datasets, where relevant variables are assumed to be recorded. Note, however, that we do *not* assume that the same variables are available at decision time for every incoming instance. Specifically, a variable may be observed in the recorded datasets used to learn the AUF probability, yet be unavailable (unmeasured/missing) in the current decision instance (see Example 2). In this sense, *learning* may rely on richer observability, whereas *decision-making* can operate under partial observability. This relaxation of observability is more realistic in practice.

## 5 Experiments

In this section, we conduct experiments to validate the utility of our measure.

**Tasks.** We simulate three synthetic tasks (Trader, Farmer, and Doctor) and a real-world case study (Bermuda). We generate 10000 samples from the underlying SEM and repeat each experiment ten times per task. Detailed settings are provided in Appendix A.

**Baselines.** We compare six methods for selecting alterations: (1) Observe, which only observes without altering variables; (2) Max-One, which selects the single variable with the highest AUF probability for alteration, as described in Equation (1); (3) Max-All, which selects all actionable variables for alteration, as described in Equation (2); (4) Corr, which sequentially selects variables with non-zero correlation and alters them to maximize the conditional AUF probability; (5) ACE, which sequentially selects variables with non-zero causal effects and alters them to maximize the interventional AUF probability; and (6) Ours, which sequentially selects variables with positive

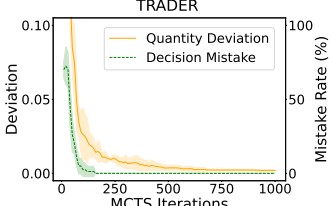 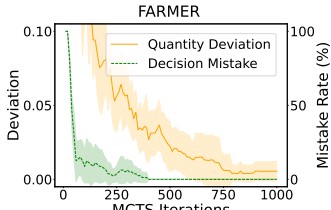 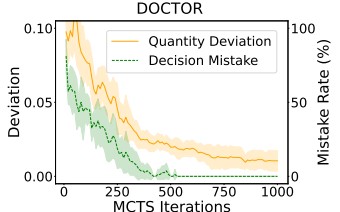

Figure 3: Convergence of influence power estimates and mistake rates (%) versus the number of MCTS iterations. The deviation of the approximated influence power from its exact value continues to decrease after the mistake rates have converged.

| TASK | OBSERVE | MAX-ONE | MAX-ALL | CORR | ACE | OURS |
|---|---|---|---|---|---|---|
| TRADER | $37.62 \pm 4.31$ | $51.01 \pm 5.13$ | $50.82 \pm 5.00$ | $47.73 \pm 6.38$ | $51.13 \pm 4.76$ | $60.94 \pm 8.43$ |
| FARMER | $10.05 \pm 2.87$ | $62.90 \pm 7.39$ | $63.18 \pm 7.64$ | $63.88 \pm 7.59$ | $62.66 \pm 7.33$ | $63.86 \pm 7.42$ |
| DOCTOR | $39.81 \pm 5.06$ | $50.76 \pm 5.50$ | $51.08 \pm 4.80$ | $51.33 \pm 5.07$ | $50.93 \pm 5.18$ | $65.32 \pm 4.12$ |

Table 1: Success rates (%) of six methods on three synthetic tasks.

| TASK | 10 | 50 | 100 | 500 | 1000 | 5000 |
|---|---|---|---|---|---|---|
| TRADER | $42.97 \pm 7.24$ | $49.45 \pm 5.12$ | $51.86 \pm 9.59$ | $57.63 \pm 6.64$ | $57.08 \pm 8.86$ | $60.34 \pm 5.64$ |
| FARMER | $19.23 \pm 11.2$ | $31.80 \pm 13.9$ | $60.49 \pm 11.9$ | $62.16 \pm 8.60$ | $63.22 \pm 8.39$ | $63.62 \pm 9.38$ |
| DOCTOR | $44.20 \pm 5.07$ | $43.18 \pm 4.86$ | $46.41 \pm 6.98$ | $64.96 \pm 5.43$ | $65.20 \pm 6.62$ | $65.72 \pm 4.50$ |

Table 2: Success rates (%) of our method with sample sizes on three synthetic tasks.

| TASK | OBSERVE | MAX-ONE | MAX-ALL | CORR | ACE | OURS |
|---|---|---|---|---|---|---|
| BERMUDA | $2.29 \pm 0.49$ | $61.99 \pm 1.43$ | $72.71 \pm 1.92$ | $19.09 \pm 5.79$ | $69.61 \pm 4.44$ | $75.16 \pm 0.89$ |

Table 3: Success rates (%) of six methods on the BERMUDA task.

influence power and alters them to maximize the MEP. Following Kocsis & Szepesvári (2006), the parameter $\alpha$ for MCTS is set to $\sqrt{2}$ by default. For fair comparison, the feasible domain for each actionable variable is set to $\{0, 1\}$ and the number of actionable variables is set to 3 for all methods in synthetic tasks. Each method is evaluated by its success rate, i.e., the frequency with which the target variable falls in the desired domain after applying the suggested alterations.

Figure 3 shows the convergence of approximating influence power. The plot depicts the deviation of the approximated value for the first actionable variable, measured as the absolute difference from the exact value. The mistake rate denotes the frequency of inconsistencies between the suggested alterations based on the approximated value and those of the exact value. In all cases, the mistake rate decreases as $T$ increases, demonstrating the effectiveness of MCTS in approximating influence power. Notably, the deviation continues to diminish after the mistake rate has converged to zero; that is, the estimate becomes decision-consistent before it fully converges numerically. This empirically supports the phenomenon discussed in Section 4.1: a coarse approximation based on a limited number of simulations can still provide useful decision guidance.

Table 1 compares our method with baselines, showing that our method outperforms existing methods in most cases. These results demonstrate the superiority of the proposed method in guiding alterations for AUF tasks. In the FARMER task, various methods perform comparably. This is because the target variable in this specific task is influenced by a single critical variable, which all five methods correctly determined. Table 2 investigates the impact of sample size on the effectiveness of our method. The performance generally improves as the sample size increases. Notably, the success rates exhibit a rapid growth initially and begin to plateau, stabilizing around 1,000 samples across the tasks.

We further validate our method on the Bermuda data to demonstrate its adaptability to real-world constraints involving non-binary variables. As shown in Table 3, our method achieves the highest success rate, outperforming the baselines. These results demonstrate that our measure is valuable and applicable in real-world scenarios with non-binary variables.

## 6 RELATED WORK

This work is grounded in the concept of *influence* and situated within the *rehearsal* paradigm (Zhou, 2022b; 2023), which advocates for proactively rehearsing future possibilities under various actions to determine alterations that influence the future target before making a final decision—an idea analogous to how human cognition prepares for future events (Driskell et al., 1994). Building upon this foundation, Qin et al. (2023) proposed a rehearsal learning approach that optimizes decisions while learning a *structural rehearsal model* capable of accommodating bi-directional interactions. Further studies have addressed issues such as non-stationarity and non-linearity in rehearsal learning (Du et al., 2024; Qin et al., 2025; Tao et al., 2025; Du et al., 2025a;b).

A substantial body of work has focused on inferring causal relations from observational data (Verma & Pearl, 1991; Cooper & Herskovits, 1992; Heckerman et al., 1995; Tian & Pearl, 2002; Shpitser & Pearl, 2006; Huang & Valtorta, 2006; Zheng et al., 2018; Lorch et al., 2021). Various measures for evaluating the strength of causal relations have also been proposed (Northcott, 2008; Janzing et al., 2013; Heskes et al., 2020; Jung et al., 2022; Janzing et al., 2024). This work primarily compares to the ACE (Rosenbaum & Rubin, 1983; Holland, 1988), as it is among the most widely used measure of causal strength in the literature; comparisons with other measures are similar and left to future work. Besides, this work differs from work based on counterfactual reasoning (Halpern, 2015; Karimi et al., 2021; Tsirtsis et al., 2021; Beckers et al., 2024). The AUF problem is inherently forward-looking, concerning planning for the future, while counterfactual reasoning typically concerns the past, asking what would have happened had we chosen differently at a point in the past (Pearl et al., 2016).

Decision-making problems have been studied from different perspectives. Reinforcement learning (RL) methods (Sutton & Barto, 2018; Lillicrap et al., 2016; Schulman et al., 2017; Haarnoja et al., 2018) often excel at learning optimal policies through extensive environmental interaction, but such interaction data can be prohibitively expensive or unavailable in many real-world settings. Moreover, unlike common RL formulations in which an agent can revisit states, real-world AUF scenarios do not permit "rewind and try again": once realized, past variables are fixed and cannot be changed. Other frameworks, such as dynamic treatment regimes (Chakraborty & Moodie, 2013; Chakraborty & Murphy, 2014; Zhang & Bareinboim, 2019; 2020), also address sequential planning, often via G-estimation or Q-learning; however, they usually enforce a strict distinction between state and action variables, as is standard in RL. In contrast, this work treats all variables uniformly as random variables; one may choose to alter a variable (set a value) or refrain from alteration (letting the variable occur naturally). Besides, causal bandits (Lattimore et al., 2016; Lee & Bareinboim, 2018) also address decision-making with causal considerations, though most existing methods require the underlying causal structure to be supplied by domain experts (Bareinboim et al., 2015; Lee & Bareinboim, 2019; 2020; Aglietti et al., 2020; Lu et al., 2020; Bilodeau et al., 2022; Wang et al., 2023; 2025; Varici et al., 2023; Branchini et al., 2023; Park et al., 2025; Park & Lee, 2025; 2026). More importantly, these approaches typically account only for the straightforward effect of intervention on the outcome, rather than the broader, holistic influence considered in this work. Furthermore, different from previous approaches based on decision networks (Howard & Matheson, 1984; Nilsson & Lauritzen, 2000; Everitt et al., 2021; Van Merwijk et al., 2022), which typically lack natural values for decision nodes, require pre-specified conditional probability distributions, and often operate under limited-memory constraints, this work acknowledges an underlying SEM as the natural generation process, enabling data-driven learning and accommodating an arbitrary number of actionable variables.

## 7 CONCLUSION

In this paper, we investigate the challenge of measuring the influence of actionable variables in the AUF problem. We present a principled quantity, termed influence power, that evaluates the degree to which altering a variable can increase the probability of a more desired future by accounting for the interplay between potential alterations and subsequent observations under the principle of maximum expected utility. Through a systematic analysis, we demonstrate an insightful distinction between influence and causation: non-ancestral variables can possess non-trivial influence power, and some causal ancestors may be ineffective or harmful to manipulate. We further develop a Monte-Carlo-based method to efficiently estimate influence power from observational data without requiring knowledge of structural equations. Experiments on synthetic and real-world tasks validate the utility of the proposed measure in guiding decisions for addressing the AUF problem.

ACKNOWLEDGMENTS

This research was supported by NSFC (62576165, 62406137) and Jiangsu Science Foundation Leading-edge Technology Program (BK20232003). Tian-Zuo Wang was supported by Xiaomi Foundation. The authors would like to thank Jia-Wei Shan and Tian Qin for their helpful comments.

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

# A  DETAILED SETTINGS

## A.1  THE SYNTHETIC TASKS

The underlying SEM for the TRADER task governs the sequence of variables $(V_1, V_2, V_3, V_4, Y)$ through the following structural equations:

$$
\begin{aligned}
V_1 &:= N_1, \\
V_2 &:= V_1 \cdot N_2 + (1 - V_1) \cdot (1 - N_2), \\
V_3 &:= V_2 \cdot N_3 + (1 - V_2) \cdot (1 - N_3), \\
V_4 &:= V_3 \cdot N_4 + (1 - V_3) \cdot (1 - N_4), \\
Y &:= V_4 \cdot (1 - V_1) + (1 - V_4) \cdot N_Y,
\end{aligned}
$$

where the noise terms follow $N_1 \sim \text{Bern}(\rho_1)$, $N_2 \sim \text{Bern}(\rho_2)$, $N_3 \sim \text{Bern}(\rho_3)$, $N_4 \sim \text{Bern}(\rho_4)$, and $N_Y \sim \text{Bern}(\rho_Y)$. To facilitate diversity across experimental trials, the task parameters are independently and uniformly sampled from the following intervals: $\rho_1 \in [0.4, 0.6]$, $\rho_i \in [0.7, 0.9]$ for $i \in \{2, 3, 4\}$, and $\rho_Y \in [0.3, 0.5]$. The variables $V_2$, $V_3$, and $V_4$ are actionable with feasible domains $\Delta_{V_2} = \Delta_{V_3} = \Delta_{V_4} = \{0, 1\}$. The variable $V_1$ is not observed at the time of decision. The desired domain is specified as $\mathcal{S} = \{1\}$. Here, $V_1$ represents the economic climate, and the chain $V_2 \to V_3 \to V_4$ models the progression from consumer demand to the final marketing strategy. The target $Y$ denotes quarterly profit, whose structural equation explicitly encodes the interaction between strategy ($V_4$) and environment ($V_1$), implying that the profitability of a specific strategy relates to the prevailing economic state. The objective is to determine the actionable variables to maximize profit.

The underlying SEM for the FARMER task governs the sequence of variables $(V_1, V_2, V_3, V_4, Y)$ thorough the following structural equations:

$$
\begin{aligned}
V_1 &:= N_1, \\
V_2 &:= (1 - V_1) \cdot N_2, \\
V_3 &:= (1 - V_2) \cdot N_3, \\
V_4 &:= (1 - V_3) \cdot N_4, \\
Y &:= V_1 \cdot V_4 \cdot N_Y,
\end{aligned}
$$

where the noise terms follow $N_1 \sim \text{Bern}(\beta_1)$, $N_2 \sim \text{Bern}(\beta_2)$, $N_3 \sim \text{Bern}(\beta_3)$, $N_4 \sim \text{Bern}(\beta_4)$, and $N_Y \sim \text{Bern}(\beta_Y)$. To facilitate diversity across experimental trials, the task parameters are independently and uniformly sampled from the following intervals: $\beta_i \in [0.7, 0.9]$ for $i \in \{1, 2, 3, 4\}$, and $\beta_Y \in [0.7, 0.9]$. The variables $V_2$, $V_3$, and $V_4$ are actionable with feasible domains $\Delta_{V_2} = \Delta_{V_3} = \Delta_{V_4} = \{0, 1\}$. The variable $V_1$ is not observed at the time of decision. The desired domain is specified as $\mathcal{S} = \{1\}$. In this context, $V_1$ represents sunlight exposure, while the chain $V_2 \to V_3 \to V_4$ models the natural water cycle affecting the soil. Specifically, intense sunlight ($V_1$) naturally reduces precipitation ($V_2$), which in turn increases evaporation ($V_3$), ultimately leading to low soil moisture ($V_4$). The target $Y$ denotes crop yield. The structural equation for $Y$ explicitly encodes the essential interaction between light ($V_1$) and water ($V_4$), implying that high productivity requires the simultaneous presence of both sunlight and adequate soil moisture. The objective is to determine the actionable variables to maximize crop yield.

The underlying SEM for the DOCTOR task governs the sequence of variables $(V_1, V_2, V_3, V_4, Y)$ thorough the following structural equations:

$$
\begin{aligned}
V_1 &:= N_1, \\
V_2 &:= N_2, \\
V_3 &:= V_1 \cdot V_2 \cdot (1 - N_3), \\
V_4 &:= N_4, \\
Y &:= V_4 \cdot (1 - V_1) + (1 - V_4) \cdot N_Y,
\end{aligned}
$$

where the noise terms follow $N_1 \sim \text{Bern}(\gamma_1)$, $N_2 \sim \text{Bern}(\gamma_2)$, $N_3 \sim \text{Bern}(\gamma_3)$, $N_4 \sim \text{Bern}(\gamma_4)$, and $N_Y \sim \text{Bern}(\gamma_Y)$. To facilitate diversity across experimental trials, the task parameters are independently and uniformly sampled from the following intervals: $\gamma_1 \in [0.4, 0.6]$, $\gamma_i \in [0.1, 0.3]$ for $i \in \{2, 3, 4\}$, and $\gamma_Y \in [0.3, 0.5]$. The variables $V_2$, $V_3$, and $V_4$ are actionable with feasible domains

$\Delta_{V_2} = \Delta_{V_3} = \Delta_{V_4} = \{0, 1\}$. The variable $V_1$ is not observed at the time of decision. The desired domain is specified as $\mathcal{S} = \{1\}$. In this context, $V_1$ represents the drug intolerance (or allergy gene), while $V_2$ represents an environmental trigger. $V_3$ denotes a symptom (e.g., a rash), which serves as a diagnostic indicator. The structural equation for $V_3$ implies that the symptom manifests primarily when both the intolerance ($V_1$) and the trigger ($V_2$) are present. $V_4$ represents the administration of a potent drug. The target $Y$ denotes patient recovery. The equation for $Y$ captures a critical medical contraindication: the drug ($V_4$) is effective for the general population ($V_1 = 0$) but is harmful or fatal to patients with the intolerance ($V_1 = 1$). The objective is to determine the actionable variables to maximize patient recovery.

## A.2    THE BERMUDA TASK

The BERMUDA case study is derived from a real-world scenario involving the management of net coral ecosystem calcification in Bermuda, where environmental variables are recorded (Aglietti et al., 2020). The sequence of variables in this task are listed as follows:

- $Light$: bottom light levels;
- $Tem$: bottom temperature;
- $Sal$: sea surface salinity;
- $DIC$: seawater dissolved inorganic carbon;
- $TA$: seawater total alkalinity;
- $\Omega_A$: seawater saturation with respect to aragonite;
- $Nut$: PC1 of $NH_4$, $NiO_2 + NiO_3$, $SiO_4$;
- $Chl\alpha$: sea surface chlorophyll-$\alpha$;
- $pH_{sw}$: seawater $pH$;
- $P_{CO_2}$: seawater $P_{CO_2}$;
- $NEC$: net ecosystem calcification.

The underlying structure governing these variables is adopted from Courtney et al. (2017) and is illustrated in Figure 4. Consistent with previous studies Aglietti et al. (2020); Qin et al. (2023), the structural equations were obtained by performing linear regression on the 50 observations provided by Andersson & Bates (2018), and there are five actionable variables including $DIC$, $TA$, $\Omega_A$, $Nut$, and $Chl\alpha$. Other variables are not observed at the time of decision. To ensure compatibility with our method, we discretize the continuous variables by dividing their value ranges into six equal-width bins. Accordingly, the feasible domains for alterations are set to $\{-2.5, -1.5, -0.5, 0.5, 1.5, 2.5\}$. The desired domain is specified as $\mathcal{S} = [1, 2]$. The objective is to determine which variables to alter in order to achieve a net ecosystem calcification ($NEC$) within the desired range, thereby promoting coral reef health and resilience.

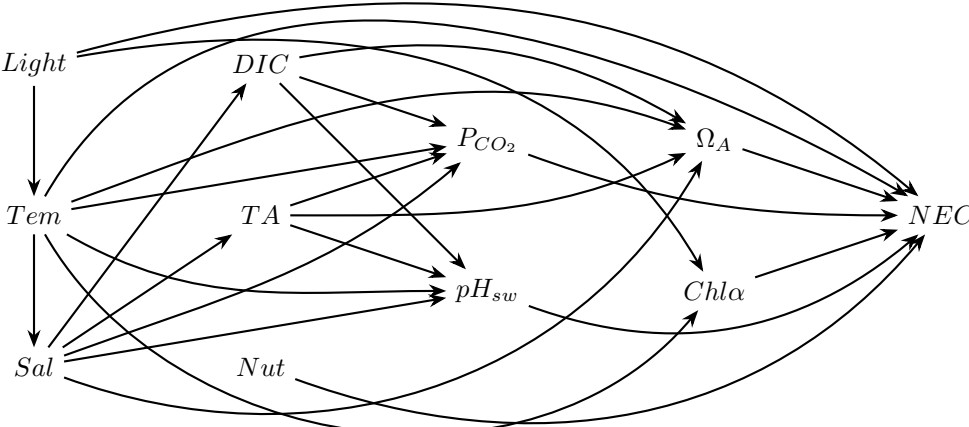

Figure 4: The underlying structure of the BERMUDA task.

# B    PROOF OF THEOREM 1

**Theorem 1.** *Let $X$ and $Y$ be two endogenous variables in an SEM. The following statements hold:*

1.    $X \in \mathrm{Anc}(Y) \;\not\Longrightarrow\; \dot{p}(X,Y) \neq 0$    *and*    $\dot{p}(X,Y) \neq 0 \;\not\Longrightarrow\; X \in \mathrm{Anc}(Y)$*;*
2.    $\tau(X,Y) \neq 0 \;\not\Longrightarrow\; \dot{p}(X,Y) \neq 0$    *and*    $\dot{p}(X,Y) \neq 0 \;\not\Longrightarrow\; \tau(X,Y) \neq 0$*;*
3.    $\tau(X,Y) \neq 0 \;\not\Longrightarrow\; \dot{p}(X,Y) \geq 0$    *and*    $X \in \mathrm{Anc}(Y) \;\not\Longrightarrow\; \dot{p}(X,Y) \geq 0$*.*

*Proof.* We prove each statement separately by constructing a counterexample.

*Statement (a)*: $X \in \mathrm{Anc}(Y) \;\not\Longrightarrow\; \dot{p}(X,Y) \neq 0$.

To show that $X \in \mathrm{Anc}(Y)$ does not imply $\dot{p}(X,Y) \neq 0$, it suffices to provide a case where a variable is an ancestor of another, yet its influence power on the latter is zero.

Consider the following SEM over the sequence of variables $(V_1, Y)$:

$$V_1 := N_1,$$
$$Y := V_1 \cdot N_Y + (1 - V_1) \cdot (1 - N_Y),$$

where $N_1 \sim \mathrm{Bern}(0.5)$, $N_Y \sim \mathrm{Bern}(0.5)$, $V_1$ is actionable with $\Delta_{V_1} = \{0,1\}$, and the desired domain for $Y$ is $\mathcal{S} = \{1\}$.

In the SEM, we have
$$V_1 \in \mathrm{Anc}(Y),$$

and
$$\begin{aligned}
\dot{p}(V_1, Y) &= \max_{v_1 \in \Delta_{V_1}} \mathcal{P}(Y = 1 | V_1 \stackrel{a}{=} v_1) - \mathbb{E}_{v_1 \sim P(V_1)} \mathcal{P}(Y = 1 | V_1 = v_1) \\
&= \max_{v_1 \in \Delta_{V_1}} P(Y = 1 | V_1 \stackrel{a}{=} v_1) - P(Y = 1) \\
&= \max\{0.5, 0.5\} - 0.5 \\
&= 0.
\end{aligned}$$

Thus, an ancestral relation in the SEM does not imply non-zero influence power.

*Statement (b)*: $\tau(X,Y) \neq 0 \;\not\Longrightarrow\; \dot{p}(X,Y) \neq 0$.

To show that $\tau(X,Y) \neq 0$ does not imply $\dot{p}(X,Y) \neq 0$, it suffices to provide a case where a variable has a non-zero average causal effect on another, yet its influence power on the latter is zero.

Consider the following SEM over the sequence of variables $(V_1, V_2, Y)$:

$$V_1 := N_1,$$
$$V_2 := V_1 \cdot N_2 + (1 - V_1) \cdot (1 - N_2),$$
$$Y := V_2 \cdot N_Y + (1 - V_2) \cdot (1 - N_Y),$$

where $N_1, N_2, N_Y \stackrel{iid}{\sim} \mathrm{Bern}(0.9)$, $V_1$ and $V_2$ are actionable with $\Delta_{V_1} = \Delta_{V_2} = \{0,1\}$, and the desired domain for $Y$ is $\mathcal{S} = \{1\}$. This SEM corresponds to Example 1 in the main text.

In the SEM, we have
$$\begin{aligned}
\tau(V_1, Y) &= P(Y = 1 | V_1 \stackrel{a}{=} 1) - P(Y = 1 | V_1 \stackrel{a}{=} 0) \\
&= 0.82 - 0.18 \\
&= 0.64,
\end{aligned}$$

and
$$\begin{aligned}
\dot{p}(V_1, Y) &= \max_{v_1 \in \Delta_{V_1}} \mathcal{P}(Y = 1 | V_1 \stackrel{a}{=} v_1) - \mathbb{E}_{v_1 \sim P(V_1)} \mathcal{P}(Y = 1 | V_1 = v_1) \\
&= \max_{v_1 \in \Delta_{V_1}} \max_{v_2 \in \Delta_{V_2}} P(Y = 1 | V_2 \stackrel{a}{=} v_2, V_1 \stackrel{a}{=} v_1) \\
&\quad - \mathbb{E}_{v_1 \sim P(V_1)} \max_{v_2 \in \Delta_{V_2}} P(Y = 1 | V_2 \stackrel{a}{=} v_2, V_1 = v_1) \\
&= \max\{0.9, 0.9\} - 0.9 \\
&= 0.
\end{aligned}$$

Thus, a non-zero average causal effect in the SEM does not imply non-zero influence power. We also note that *Statement (b)* implies *Statement (a)*, as $\tau(X, Y) \neq 0$ implies $X \in \text{Anc}(Y)$.

*Statement (c)*: $\dot{p}(X, Y) \neq 0 \not\Rightarrow \tau(X, Y) \neq 0$.

To show that $\dot{p}(X, Y) \neq 0$ does not imply $\tau(X, Y) \neq 0$, it suffices to provide a case where a variable has non-zero influence power on another, yet its average causal effect on the latter is zero.

Consider the following SEM over the sequence of variables $(V_1, V_2, Y)$:

$$
\begin{aligned}
V_1 &:= N_1, \\
V_2 &:= (1 - V_1) \cdot N_2, \\
Y &:= V_1 \cdot V_2 \cdot N_Y,
\end{aligned}
$$

where $N_1, N_2, N_Y \overset{iid}{\sim} \text{Bern}(0.5)$, $V_1$ and $V_2$ are actionable with $\Delta_{V_1} = \Delta_{V_2} = \{0, 1\}$, and the desired domain for $Y$ is $\mathcal{S} = \{1\}$. This SEM corresponds to Example 3 in the main text.

In the SEM, we have

$$
\begin{aligned}
\tau(V_1, Y) &= P(Y = 1 | V_1 \overset{a}{=} 1) - P(Y = 1 | V_1 \overset{a}{=} 0) \\
&= 0.82 - 0.18 \\
&= 0.64,
\end{aligned}
$$

and

$$
\begin{aligned}
\dot{p}(V_1, Y) &= \max_{v_1 \in \Delta_{V_1}} \mathcal{P}(Y = 1 | V_1 \overset{a}{=} v_1) - \mathbb{E}_{v_1 \sim P(V_1)} \mathcal{P}(Y = 1 | V_1 = v_1) \\
&= \max_{v_1 \in \Delta_{V_1}} \max_{v_2 \in \Delta_{V_2}} P(Y = 1 | V_2 \overset{a}{=} v_2, V_1 \overset{a}{=} v_1) \\
&\quad - \mathbb{E}_{v_1 \sim P(V_1)} \max_{v_2 \in \Delta_{V_2}} P(Y = 1 | V_2 \overset{a}{=} v_2, V_1 = v_1) \\
&= \max\{0, 0.5\} - 0.25 \\
&= 0.25.
\end{aligned}
$$

Thus, non-zero influence power does not imply a non-zero average causal effect in the SEM.

*Statement (d)*: $\dot{p}(X, Y) \neq 0 \not\Rightarrow X \in \text{Anc}(Y)$.

To show that $\dot{p}(X, Y) \neq 0$ does not imply $X \in \text{Anc}(Y)$, it suffices to provide a case where a variable has non-zero influence power on another, yet it is not an ancestor of the latter.

Consider the following SEM over the sequence of variables $(V_1, V_2, V_3, V_4, Y)$:

$$
\begin{aligned}
V_1 &:= N_1, \\
V_2 &:= N_2, \\
V_3 &:= V_1 \cdot V_2 \cdot (1 - N_3), \\
V_4 &:= N_4, \\
Y &:= V_4 \cdot (1 - V_1) + (1 - V_4) \cdot N_Y,
\end{aligned}
$$

where $N_1 \sim \text{Bern}(0.5)$, $N_2, N_3, N_4 \overset{iid}{\sim} \text{Bern}(0.1)$, $N_Y \sim \text{Bern}(0.4)$, $V_2, V_3$, and $V_4$ are actionable with $\Delta_{V_2} = \Delta_{V_3} = \Delta_{V_4} = \{0, 1\}$, and the desired domain for $Y$ is $\mathcal{S} = \{1\}$. This SEM corresponds to Example 2 in the main text.

In the SEM, we have

$$
V_2 \notin \text{Anc}(Y),
$$

and

$$\dot{p}(V_2, Y) = \max_{v_2 \in \Delta_{V_2}} \mathcal{P}(Y = 1 | V_2 \overset{a}{=} v_2) - \mathbb{E}_{v_2 \sim P(V_2)} \mathcal{P}(Y = 1 | V_2 = v_2)$$

$$= \max_{v_2 \in \Delta_{V_2}} \max \Big\{ \max_{v_3 \in \Delta_{V_3}} \mathcal{P}(Y = 1 | V_3 \overset{a}{=} v_3, V_2 \overset{a}{=} v_2),$$

$$\mathbb{E}_{v_3 \sim P(V_3 | V_2 \overset{a}{=} v_2)} \mathcal{P}(Y = 1 | V_3 = v_3, V_2 \overset{a}{=} v_2) \Big\}$$

$$- \mathbb{E}_{v_2 \sim P(V_2)} \max \Big\{ \max_{v_3 \in \Delta_{V_3}} \mathcal{P}(Y = 1 | V_3 \overset{a}{=} v_3, V_2 = v_2),$$

$$\mathbb{E}_{v_3 \sim P(V_3 | V_2 = v_2)} \mathcal{P}(Y = 1 | V_3 = v_3, V_2 = v_2) \Big\}$$

$$= \max_{v_2 \in \Delta_{V_2}} \mathbb{E}_{v_3 \sim P(V_3 | V_2 \overset{a}{=} v_2)} \mathcal{P}(Y = 1 | V_3 = v_3, V_2 \overset{a}{=} v_2)$$

$$- \mathbb{E}_{v_2 \sim P(V_2)} \mathbb{E}_{v_3 \sim P(V_3 | V_2 = v_2)} \mathcal{P}(Y = 1 | V_3 = v_3, V_2 = v_2)$$

$$= \max_{v_2 \in \Delta_{V_2}} \mathbb{E}_{v_3 \sim P(V_3 | V_2 \overset{a}{=} v_2)} \max_{v_4 \in \Delta_{V_4}} P(Y = 1 | V_4 \overset{a}{=} v_4, V_3 = v_3, V_2 \overset{a}{=} v_2)$$

$$- \mathbb{E}_{v_2 \sim P(V_2)} \mathbb{E}_{v_3 \sim P(V_3 | V_2 = v_2)} \max_{v_4 \in \Delta_{V_4}} P(Y = 1 | V_4 \overset{a}{=} v_4, V_3 = v_3, V_2 = v_2)$$

$$= 0.68 - 0.518$$

$$= 0.162.$$

Thus, non-zero influence power does not imply an ancestral relation in the SEM.
We also note that *Statement (d)* implies *Statement (c)*, as $X \notin \mathrm{Anc}(Y)$ implies $\tau(X, Y) = 0$.

*Statement (e)*: $X \in \mathrm{Anc}(Y) \not\Longrightarrow \dot{p}(X, Y) \geq 0$.

To show that $X \in \mathrm{Anc}(Y)$ does not imply $\dot{p}(X, Y) \geq 0$, it suffices to provide a case where a variable is an ancestor of another, yet its influence power on the latter is negative.

Consider the following SEM over the sequence of variables $(V_1, V_2, Y)$:

$$V_1 := N_1,$$
$$V_2 := (1 - V_1) \cdot N_2,$$
$$Y := (V_1 \oplus V_2) \cdot N_Y,$$

where $N_1 \sim \mathrm{Bern}(0.5)$, $N_2, N_Y \overset{iid}{\sim} \mathrm{Bern}(0.8)$, $V_2$ is actionable with $\Delta_{V_2} = \{0, 1\}$, and the desired domain for $Y$ is $\mathcal{S} = \{1\}$.

In the SEM, we have
$$V_2 \in \mathrm{Anc}(Y),$$
and
$$\dot{p}(V_2, Y) = \max_{v_2 \in \Delta_{V_2}} \mathcal{P}(Y = 1 | V_2 \overset{a}{=} v_2) - \mathbb{E}_{v_2 \sim P(V_2)} \mathcal{P}(Y = 1 | V_2 = v_2)$$
$$= \max_{v_2 \in \Delta_{V_2}} P(Y = 1 | V_2 \overset{a}{=} v_2) - P(Y = 1)$$
$$= \max\{0.4, 0.4\} - 0.72$$
$$= -0.32.$$

Thus, an ancestral relation in the SEM does not imply non-negative influence power.

*Statement (f)*: $\tau(X, Y) \neq 0 \not\Longrightarrow \dot{p}(X, Y) \geq 0$.

To show that $\tau(X, Y) \neq 0$ does not imply $\dot{p}(X, Y) \geq 0$, it suffices to provide a case where a variable has a non-zero average causal effect on another, yet its influence power on the latter is negative.

Consider the following SEM over the sequence of variables $(V_1, V_2, V_3, Y)$:

$$V_1 := N_1,$$
$$V_2 := V_1 \cdot N_2 + (1 - V_1) \cdot (1 - N_2),$$
$$V_3 := V_2 \cdot N_3 + (1 - V_2) \cdot (1 - N_3),$$
$$Y := V_3 \cdot (1 - V_1) + (1 - V_3) \cdot N_Y,$$

where $N_1 \sim \text{Bern}(0.5)$, $N_2, N_3 \overset{iid}{\sim} \text{Bern}(0.9)$, $N_Y \sim \text{Bern}(0.4)$, $V_2$ and $V_3$ are actionable with $\Delta_{V_2} = \{0, 1\}$ and $\Delta_{V_3} = \{0, 1\}$, and the desired domain for $Y$ is $\mathcal{S} = \{1\}$. This SEM corresponds to Example 4 in the main text.

In the SEM, we have

$$
\begin{aligned}
\tau(V_2, Y) &= P(Y = 1 | V_2 \overset{a}{=} 1) - P(Y = 1 | V_2 \overset{a}{=} 0) \\
&= 0.49 - 0.41 \\
&= 0.08,
\end{aligned}
$$

and

$$
\begin{aligned}
\dot{p}(V_2, Y) &= \max_{v_2 \in \Delta_{V_2}} \mathcal{P}(Y = 1 | V_2 \overset{a}{=} v_2) - \mathbb{E}_{v_2 \sim P(V_2)} \mathcal{P}(Y = 1 | V_2 = v_2) \\
&= \max_{v_2 \in \Delta_{V_2}} \max_{v_3 \in \Delta_{V_3}} P(Y = 1 | V_3 \overset{a}{=} v_3, V_2 \overset{a}{=} v_2) \\
&\quad - \mathbb{E}_{v_2 \sim P(V_2)} \max_{v_3 \in \Delta_{V_3}} P(Y = 1 | V_3 \overset{a}{=} v_3, V_2 = v_2) \\
&= \max\{0.5, 0.5\} - 0.65 \\
&= -0.15.
\end{aligned}
$$

Thus, a non-zero average causal effect in the SEM does not imply non-negative influence power. We also note that *Statement (f)* implies *Statement (e)*, as $\tau(X, Y) \neq 0$ implies $X \in \text{Anc}(Y)$. □

## C    PROOF OF PROPOSITION 1

**Proposition 1.** *Assume causal sufficiency, i.e., the joint distribution $P(\mathbf{V}, Y)$ is induced by an acyclic SEM $\mathcal{M}$ with independent noises. Also assume positivity, i.e., $P(V_i | \text{PA}_i) > 0$ for all $1 \leq i \leq d$ on the support of $P$. Then, the expression in Equation (11) is consistent with the joint probability dictated by the SEM under the alteration of $\mathbf{A}$. Furthermore, the expression in Equation (12) is consistent with the AUF probability dictated by the SEM under the alteration of $\mathbf{A}$ and the observation of $\mathbf{O}$.*

*Proof.* Recall from Equation (11), the joint distribution conditioned on the alteration set $\hat{\mathbf{A}}$ is expressed as $P(\mathbf{V} | \hat{\mathbf{A}}) = \prod_{V_i \in \mathbf{A}} \delta(V_i) \prod_{V_i \in \mathbf{V} \setminus \mathbf{A}} P(V_i | V_1, \ldots, V_{i-1})$. As the sequence is topologically consistent with the underlying SEM, and the SEM is assumed to be acyclic, the value of each variable $V_i$ depends solely on its parents $\text{PA}_i$. Consequently, $P(V_i | V_1, \ldots, V_{i-1}) = P(V_i | \text{PA}_i)$. Substituting this back into the product shows that $P(\mathbf{V} | \hat{\mathbf{A}}) = \prod_{V_i \in \mathbf{A}} \delta(V_i) \prod_{V_i \in \mathbf{V} \setminus \mathbf{A}} P(V_i | \text{PA}_i)$. By invoking the manipulation theorem (i.e., Theorem 3.6 in Spirtes et al. (2000)), we have that $P(\mathbf{V} | \hat{\mathbf{A}})$ is exactly the probability of $\mathbf{V}$ under the alteration of $\mathbf{A}$. Moreover, the quantity $P(Y \in \mathcal{S} | \hat{\mathbf{A}}, \mathbf{O})$ in Equation (12) is fully determined by $P(\mathbf{V} | \hat{\mathbf{A}})$, and therefore Equation (12) indeed gives to the true AUF probability dictated by the underlying SEM. □

