# OpenReview forum: "On Measuring Influence in Avoiding Undesired Future"
_ICLR.cc/2026/Conference — ICLR 2026 Poster_

### Official Review · Reviewer_rkWZ · 2025-10-31

**Soundness:** 4
**Presentation:** 4
**Contribution:** 3
**Rating:** 8
**Confidence:** 2

**Summary:**

The paper introduces the notion of influence power, a measure of the potential
importance of a variable for preventing a system from leading to an undesirable
outcome. This notion is related to, but is shown to differ, to that of average
causal effects. The paper also introduces a UCT-based Monte-Carlo Tree Search
estimator (from noiseless observational data). Theoretical support is provided
for the consistency of the estimator in the unconfounded case, as well as
empirical support on three toy tasks with noiseless data.

**Strengths:**

**Originality**: to the best of my knowledge, the contribution is novel. It
is definitely related to work in utility maximization and possibly algorithmic
recourse, but it is also different enough -- as clarified by the authors.

[**Q**] Could you please clarify the points of difference with works in causal
algorithmic recourse? E.g., the works by Karimi and colleagues.  I'm sure this
would help the readers.

**Quality**:

**Clarity**: The text is well written and easy to follow.  All key definitions
and arguments are accompanied by illustrative examples, which help a lot.

**Significance**: I am not the most well versed in algorithmic decision making,
but I think this paper provides a useful contribution mixing elements from causality and decision making, and it could open the door to further research.

I'm curious to know what the other reviewers think.

**Weaknesses:**

Focus on constructive and actionable insights on how the work could improve
towards its stated goals.

**Clarity**: No major complaints on my end, but I did notice a handful of small
linguistic idiosyncrasies, such as:

- Title: I'm not a native English speaker, but the sentence "avoiding
  undersired future" sounds off to me. What about "Avoiding undesirable future
  events"? This is how it's written in the abstract and it works just fine.

- "alterable variable" also sounds off to me -- what about "actionable variable"?

- Section 2, Notation: I don't think the notation {\cal M}_{V_i} used
  elsewhere in the text, it can probably be removed. Especially because it
  looks incorrect: I suspect it should be {\cal M}_{v_i} instead (lower case,
  constant).

- Section 2, Problem Definition: no need to repeat that \Delta_{V_i} is the
  alterable domain. This was already introduced previously.

- Section 3.2: "an alterable variable are worth" -> is worth.

- Section 3.3: "Also, a variable such as..." - why "also"? I'd drop it.

- Above Example 4: "it is not a causal" -> "not being a causal".

A minor issue is it's not immediately obvious to see the difference between
the two main terms in the main equation (at the end of p 3): one contains an
intervention, the other an observation, but they are marked as "a" and "o",
which are quite difficult to recognize as different. Perhaps use color to
help the reader? The equation is much easier to understand once one spots this
otherwise tiny difference ;-)

**Significance**: [**Q**] It'd be good to report the runtime difference between the
three algorithms (the baselines and the MCTS estimator). The results indicate
the estimator outperforms the baselines, which is good, but without
understanding the its computational cost it's difficult to gauge whether the
benefits are worth it. This is my only major complaing with the work.

[**Q**] It'd also be good to clarify from the get go - as soon as the introduction -
that Proposition 1 works only for unconfounded models ("independent background
noises"), which can be a strong assumption in practice, for clarity.

**Questions:**

I'd appreciate if the authors could comment on the points I've marked as [**Q**] above.

---

> ### Author Response · Authors · 2025-11-26
> **Response to Reviewer rkWZ (Part 1)**
>
> Thank you for recognizing our work and the great comments. We hope the following responses can fully address your concerns; please let us know if any points remain unclear.
>
> ---
>
> > [Q] Could you please clarify the points of difference with works in causal algorithmic recourse? E.g., the works by Karimi and colleagues. I'm sure this would help the readers.
>
> Causal algorithmic recourse [1] relies on counterfactuals, as it extends the setting of counterfactual explanations: given a factual instance $\boldsymbol{x}$, the goal is to find a counterfactual instance $\boldsymbol{x}'$ such that the prediction changes (i.e., $h(\boldsymbol{x}) \neq h(\boldsymbol{x}')$). In contrast, our framework addresses a forward-looking problem where factual values that have already occurred are immutable (i.e., the past cannot be changed; only future variables are actionable). Consequently,  the proposed quantity is computed using interventional probabilities rather than counterfactual probabilities.
>
> ---
>
>
> > the sentence "avoiding undersired future" sounds off to me. What about "Avoiding undesirable future events"? This is how it's written in the abstract and it works just fine.
>
> Thank you very much for the suggestion. We agree that "avoiding undesirable future events" is a clear and standard phrasing, and it works well in the abstract where the context explicitly refers to future events. In the sentence under discussion, however, our intention was to emphasize the notion of an undesired future as a general state, rather than specific events. Therefore, "avoiding undesired future" was chosen to capture this broader and more conceptual meaning.
>
> ---
>
> > "alterable variable" also sounds off to me -- what about "actionable variable"?
> > Section 2, Notation: I don't think the notation {\cal M}{V_i} used elsewhere in the text, it can probably be removed. Especially because it looks incorrect: I suspect it should be {\cal M}{v_i} instead (lower case, constant).
> > Section 2, Problem Definition: no need to repeat that \Delta_{V_i} is the alterable domain. This was already introduced previously.
> >
> > Section 3.2: "an alterable variable are worth" -> is worth.
> >
> > Section 3.3: "Also, a variable such as..." - why "also"? I'd drop it.
> >
> > Above Example 4: "it is not a causal" -> "not being a causal".
>
> We thank the reviewer for the detailed feedback. We have incorporated all these suggestions and polished the writing in the revised manuscript.

---

> ### Author Response · Authors · 2025-11-26
> **Response to Reviewer rkWZ (Part 2)**
>
> > A minor issue is it's not immediately obvious to see the difference between the two main terms in the main equation (at the end of p 3): one contains an intervention, the other an observation, but they are marked as "a" and "o", which are quite difficult to recognize as different. Perhaps use color to help the reader? The equation is much easier to understand once one spots this otherwise tiny difference ;-)
> >
>
> Thanks for pointing this out. To avoid confusion and improve readability, we have replaced the notations $V_i \overset{\textit{a}}{=} v_i$ and $V_i \overset{\textit{o}}{=} v_i$ with $V_i \coloneq v_i$ (for alteration) and $V_i = v_i$ (for observation), respectively, throughout the revised manuscript.
>
> ---
>
> > [Q] It'd be good to report the runtime difference between the three algorithms (the baselines and the MCTS estimator). The results indicate the estimator outperforms the baselines, which is good, but without understanding the its computational cost it's difficult to gauge whether the benefits are worth it. This is my only major complaing with the work.
>
> Thanks for the suggestion. We have conducted a runtime analysis comparing our approach against the baselines. The average runtime per single execution (in seconds) is reported below:
>
> | **Task** | **Observe** | Max-One | Max-All | **MIS** | **VoC** | **Ours** |
> | -------- | ----------- | ------- | ------- | ------- | ------- | -------- |
> | Trader   | 0.2968      | 0.9600  | 1.4139  | 1.0192  | 2.0904  | 5.3361   |
> | Farmer   | 0.2124      | 0.8384  | 1.3025  | 0.8945  | 1.8140  | 3.3566   |
> | Doctor   | 0.2679      | 0.8816  | 1.3366  | 0.9206  | 2.0073  | 5.2135   |
>
> While our method incurs a higher computational cost due to the MCTS exploration, the runtime remains within a feasible range for decision-making.
>
> ---
>
> > [Q] It'd also be good to clarify from the get go - as soon as the introduction - that Proposition 1 works only for unconfounded models ("independent background noises"), which can be a strong assumption in practice, for clarity.
>
> Thanks for the suggestion. We have explicitly clarified the assumption in the introduction of the revised manuscript.
>
> ---
>
> **Reference**
>
> [1] Karimi, Amir-Hossein, Bernhard Schölkopf, and Isabel Valera. "Algorithmic recourse: from counterfactual explanations to interventions." Proceedings of the 2021 ACM conference on fairness, accountability, and transparency. 2021.

---

### Official Review · Reviewer_NVBs · 2025-10-31

**Soundness:** 3
**Presentation:** 3
**Contribution:** 3
**Rating:** 6
**Confidence:** 4

**Summary:**

The authors present a novel approach to the AUF (avoiding undesirable future) problem
which improves on SOTA (one-alteration and all-alteration). Their approach can also capture negative
influence, ie changes which are strictly bad with respect to a given target. They use a Monte Carlo
algorithm to make estimating their measure more efficient (than brute force). They evaluate their
method on three standard(?) problems and show that their measure allows them to compute
interventions which increase the chances of the model reaching the desired target region.

**Strengths:**

I quite like this paper, I'd give it a weak accept. It's far from perfect, the novelty (while
there) is not earth-shattering and most of the worked
examples are little more than toys, but I enjoyed reading it.

**Weaknesses:**

I quite like this paper, I'd give it a weak accept. It's far from perfect, the novelty (while
there) is not earth-shattering and most of the worked
examples are little more than toys, but I enjoyed reading it. See the questions below.

**Questions:**

(numbers are line numbers)

089 how much is nearly negligible?
103 it is assumed that all variables are alterable? How strong is this assumption?
- it feels like a very strong assumption to me, as any variable on the causal path (or off of it)
  can be directly manipulated. This certainly does not reflect real world limitations.

104 "desired region" feels a little vague. S is a subset of the possible values of Y?
107 "as much as possible" is also rather vague. Is there an implicit threshold or ratio here?

I don't think that eq 2 implies that all variables *must* be altered, only that they can be altered.
After all, the equation simply states that we must go over all variables and set their values, but
it does not state that the values must be **different** from observed. So the criticism on l.139 does not hold.

133 do not use contractions

155 I don't like the way this is presented. k = d comes out of nowhere, and the MEP and AUF
equations appear to be identical at this point. It's only later that it becomes (slightly) more
obvious that this is deliberate.

158 eq 3 is not well formatted. =o, unlike =a, has not been previously defined. Presumably it means
setting V_k to its actual value in the context. But wait, we get the definition on l.164. This is
backwards and very hard to read. Why not use <- for alteration, and = for observation?

173 Def 1: clearly related to average causal effect. Is it a generalisation?

211 causality need not be transitive. See Halpern for a detailed discussion.
It is also not that surprising that a alterable variable be non-influential. After all, the variable
may cause an exact value of Y in S, but not have sufficient influence to move out of S.

200 notation for \tau is really buried in the text here.

226 this is a slightly forced example, as X and Z are essentially independent, they are both on the
causal path, but X is not a cause. However, it's good to see that IP can detect this.

245 here we have the problem of talking about negligible ACE, as 0.08 is surely very low. Is this
negligible (see comment above)?

262 again, not a surprising result. And the influence (271) is small.

274 small note, but it should read "considering altering", not "considering to alter"

278 "despite it not being a causal ancestor"

Example 4. I do not understand your explanation. It is not intuitive (to me) and seems quite
an important point. This needs clarification.

Eq (9) is the \delta embedded in the equation standard notation for something?
What is A hat? The MC approximation of A at any point?

420 "repeat experiment _with_ ten times"

421 I looked at Appendix A. I'm slightly disappointed by how small/simple the examples are. I wonder
how well the MC sampling approach would work on bigger/more difficult models. Moreover, they are
largely identical to your examples. I think you could just merge this together, name the examples
appropriately, highlight differences if they exist, and remove the appendix. It would make it much
easier to read.

438 "demosntrating"

Figure 2 is very small and quite difficult to read. Please replot or make larger for camera ready.

Table 1 does not show how much work is required for your MC method. Your results are already better
at T=10. I'd like an idea of how much comparable work is performed by Max-One and Max-All.
I'm assuming (because it's not in the comment at least) that these are mean values. While I'm not in
favour of unnecessary statistical analysis, some might be appropriate here. Perhaps a box plot might
be more revealing here, as the standard deviations are quite large for all the tools, and this is
someone obscured by presenting the data as a table.

The results seems to follow a logaritmic curve... why is this? Is this a limitation of the the MC
method? After all, it should be possible, if all variables in V are alterable, to always guarantee Y
in S? Or is this a result of the fact that you cannot alter variables before some point d?

Conclusion section. "intriguing possibility..." but I do not understand why this happens, and your
intuition above is not (to me) intuitive. I could perhaps see the sharing of information via mutual
information at play, but this is non-directional. Is this not instead revealing a limitation in
(Pearl's conception of) SCMs based on probabilities? It's probably always possible to construct
these pathological examples (because they are divorced from real data producing systems) but do they
really tell us anything interesting, other than to look at for these potential errors?

_Questions_:
  * distance of alterations from outcome Y. Are they usually proximate or far away?
  * are interventions consecutive or disjoint in general?
  * are there any patterns/implications in distance from Y?

  * frequency of achieving S in your simulations?
      + S is always limited to one outcome I think, so the entire thing is boolean. Will this scale
    to non-boolean settings.
  * size of alterations vs frequency of S?

---

> ### Author Response · Authors · 2025-11-26
> **Response to Reviewer NVBs (Part 1)**
>
> Thank you for the thoughtful feedback. We hope the following responses can fully address your concerns; please let us know if any points remain unclear.
>
> ---
>
> > 089 how much is nearly negligible?
>
> While defining "nearly" can be subjective, we can safely state that an average causal effect (ACE) of zero is negligible. To eliminate this ambiguity, we have revised the phrasing in the manuscript accordingly.
>
> > 103 it is assumed that all variables are alterable? How strong is this assumption?
>
> We do not assume that all variables are alterable; actually, our framework explicitly accommodates unalterable variables. Specifically, each variable $V_i$ is associated with a feasible alterable domain $\Delta_{V_i}$. If a variable cannot be altered, its corresponding domain $\Delta_{V_i}$ is defined as an empty set. This allows for a flexible modeling of real-world constraints.
>
> > 104 "desired region" feels a little vague. S is a subset of the possible values of Y?
>
> Yes, your interpretation is correct: $\mathcal{S}$ represents a specific subset of the possible values of $Y$. Referring to $\mathcal{S}$as the "desired domain" might be more precise. Thank you for pointing this out.
>
> > 107 "as much as possible" is also rather vague. Is there an implicit threshold or ratio here?
>
> This means that the goal is to maximize the probability of $Y \in \mathcal{S}$. There is no implicit threshold or ratio in the problem definition.
>
> > I don't think that eq 2 implies that all variables must be altered, only that they can be altered. After all, the equation simply states that we must go over all variables and set their values, but it does not state that the values must be different from observed. So the criticism on l.139 does not hold.
>
> Thanks for pointing this out. A more accurate description would be that Eq. 2 is capable of altering all variables, rather than indiscriminately altering all variables. We have refined the criticism of Eq. 2 in the revised manuscript to reflect this nuance accurately.
>
>
> > 155 I don't like the way this is presented. k = d comes out of nowhere, and the MEP and AUF equations appear to be identical at this point. It's only later that it becomes (slightly) more obvious that this is deliberate.
>
> Thank you for pointing this out. We have reorganized the presentation in the revised manuscript. We now state the general case where $0 < k < d$ first, followed by the specific case where $k=d$. This adjustment improves the logical flow and clarifies the distinction between the MEP and AUF equations.
>
> > 158 eq 3 is not well formatted. =o, unlike =a, has not been previously defined. Presumably it means setting V_k to its actual value in the context. But wait, we get the definition on l.164. This is backwards and very hard to read. Why not use <- for alteration, and = for observation?
>
> We appreciate the suggestion regarding notation clarity. To align with standard conventions and improve readability, we have replaced the notations $V_i \overset{\textit{a}}{=} v_i$ and $V_i \overset{\textit{o}}{=} v_i$ with $V_i \coloneq v_i$ (for alteration) and $V_i = v_i$ (for observation), respectively, throughout the revised manuscript.
>
> > 173 Def 1: clearly related to average causal effect. Is it a generalisation?
>
> This is an interesting question. To shed light on the relationship, consider a simplified case with two variables $X$ and $Y$. Let the desired domain be $\mathcal{S}=\{1\}$. Denote $P(Y=1 \mid X \coloneq 1)$, $P(Y=1 \mid X \coloneq 0)$, and $P(Y=1)$ as $p_1$, $p_0$, and $p$, respectively. The average causal effect is given as $\tau(X,Y) = p_1 - p_0$. The influence power is derived as $\dot{p}(X, Y) = \max\{p_1, p_0\} - p$. Using the identity $\max\{a,b\} = \frac{a+b+|a-b|}{2}$, we can rewrite the influence power as: $ \dot{p}(X, Y) = \frac{p_1 + p_0 + |\tau(X,Y)|}{2} - p $. Thus, the influence power is closely related to the average causal effect. It can be interpreted as an extension of the average causal effect in this case, though it does not strictly generalize it.
>
> > 245 here we have the problem of talking about negligible ACE, as 0.08 is surely very low. Is this negligible (see comment above)?
>
> We would like to clarify that 0.08 is not negligible. While it may be difficult to define an exact threshold for what constitutes "nearly negligible," we can confidently state that a value of 0 is negligible.
>
> > 262 again, not a surprising result. And the influence (271) is small.
>
> While an influence power of 0.09 is small, we would like to clarify that it is not negligible. In practical terms, this value represents an increase of 9% in the maximum expected probability of $Y\in\mathcal{S}$ achievable through alteration.

---

> ### Author Response · Authors · 2025-11-26
> **Response to Reviewer NVBs (Part 2)**
>
> > Example 4. I do not understand your explanation. It is not intuitive (to me) and seems quite an important point. This needs clarification.
>
> To provide better intuition, let us ground the variables in Example 4 in a realistic medical scenario: Consider a patient visiting a doctor. Let $U$, $W$, $X$, $Z$, and $Y$ denote the allergy gene, skin test, skin response, drug injection, and recovery, respectively. Performing a skin test ($W$) has no direct therapeutic effect, so the average causal effect of $W$ on $Y$ is zero. However, performing the skin test ($W$) is crucial because it informs the doctor's decision on whether to administer the drug ($Z$), which indirectly influences the patient’s recovery ($Y$). If the skin test result is positive ($R=1$ after setting $W\coloneq 1$), the doctor might decide not to administer the drug ($Z\coloneq 0$), as this would maximize the probability of recovery ($Y=1$). This scenario illustrates that, although $W$ does not intrinsically cause $Y$, altering $W$ can still be useful and indirectly influence $Y$. We have added this concrete context to the revised manuscript to improve clarity.
>
> > Eq (9) is the \delta embedded in the equation standard notation for something? What is A hat?
>
> Thanks for pointing this out. In Eq (9), $\delta(V_i)$ is a abbreviation of the Dirac delta function $\delta(V_i - v_i)$, which equals to $1$ if $V_i = v_i$ and $0$ if $V_i \neq v_i$, where $v_i$ is any value within the alteration domain $\Delta_{V_i}$. The term $\hat{\mathbf{A}}$ denotes the alteration of the set $\mathbf{A}$, implying that every variable $V_i \in \mathbf{A}$ is altered to some value $v_i$. We have clarified these notations in the revised manuscript.
>
> > 421 I looked at Appendix A. I'm slightly disappointed by how small/simple the examples are. I wonder how well the MC sampling approach would work on bigger/more difficult models. Moreover, they are largely identical to your examples. I think you could just merge this together, name the examples appropriately, highlight differences if they exist, and remove the appendix. It would make it much easier to read.
>
> Thanks for the suggestion. The detailed synthetic tasks are actually more diverse than the examples in the main text. We have consolidated Appendix A and included new experimental results on a more difficult task with non-binary variables in the revised manuscript.
>
> > The results seems to follow a logaritmic curve... why is this? Is this a limitation of the the MC method? After all, it should be possible, if all variables in V are alterable, to always guarantee Y in S? Or is this a result of the fact that you cannot alter variables before some point d?
>
> The logarithmic-like convergence is a characteristic pattern of Monte-Carlo simulations. The simulation initially identifies high-impact alterations easily (resulting in a steep initial slope), but as the simulation progresses, finding further marginal improvements requires significantly more samples, causing the curve to flatten. This reflects the natural diminishing returns of random sampling in complex spaces.
>
> Regarding the guarantee of $Y \in \mathcal{S}$: Even if all variables are alterable and set optimally, we cannot guarantee the outcome with probability 1 because the target variable $Y$ is often inherently stochastic. For example, if $Y \sim \text{Bern}(0.5 \cdot \prod_{i=1}^d V_i)$, the target region is $\mathcal{S}=\{1\}$, and all feasible domains are $\{0,1\}$, the maximum achievable probability of $Y \in \mathcal{S}$ is only 50%, even if we set every variable $V_i$ to 1.

---

> ### Author Response · Authors · 2025-11-26
> **Response to Reviewer NVBs (Part 3)**
>
> > Conclusion section. "intriguing possibility..." but I do not understand why this happens, and your intuition above is not (to me) intuitive. I could perhaps see the sharing of information via mutual information at play, but this is non-directional. Is this not instead revealing a limitation in (Pearl's conception of) SCMs based on probabilities? It's probably always possible to construct these pathological examples (because they are divorced from real data producing systems) but do they really tell us anything interesting, other than to look at for these potential errors?
>
> We appreciate this opportunity to clarify the intuition. To illustrate, let us revisit the medical example: Consider a skin test ($W$) and a drug injection ($Z$) for patient recovery ($Y$).
>
> - Natural Mechanism: Physically, the skin test does not cure the patient (a randomized controlled trial would show no causal effect of $W$ on $Y$). In a standard SCM describing this natural data-generating process, there is no causal path from the skin test to recovery (i.e., $W \to \text{Skin Response}$, $Z \to Y$, but no arrow from $\text{Skin Response} \to Z$).
> - Decision Process: However, for a decision-maker (the doctor), administering the skin test is highly beneficial. The result of the test provides critical information that allows the doctor to adjust the subsequent decision on drug injection ($Z$), thereby optimizing the target $Y$.
>
> Therefore, the influence of the skin test on recovery is manifest in the context of the decision task, even though it is absent in the physical causal structure. This distinction might not reflect a shortcoming of Pearl’s SCM framework; rather, it highlights the fundamental difference between the natural data-generating process (where the test does not physically cause recovery) and the decision process (where the test indirectly influences recovery by informing the agent's actions).
>
> > distance of alterations from outcome Y. Are they usually proximate or far away? Are there any patterns/implications in distance from Y?
>
> Intuitively, one might assume that altering variables "close" to $Y$ would be more beneficial. While this holds in some cases, it is not a general rule, and topological distance is not a reliable proxy for influence power. The specific functional relationships within the SCM are far more determinant. Consider the following counter-example: Let $V_1 \sim \text{Bern}(0.5)$, $V_2 \coloneqq 1-V_1$, and $Y \coloneqq V_1 + (1 - V_1) \cdot V_2 \cdot N$, where $N \sim \text{Bern}(0.5)$ is background noise, and the desired domain is $\mathcal{S}=\{1\}$.
>
> - Distance: $V_2$ is closer to $Y$ than $V_1$ in the sequence.
> - Influence: Altering $V_2$ yields no positive gain over the natural state ($P(Y=1 \mid V_2 \coloneq 1) = 0.75$, which is equal to the natural probability $P(Y=1)=0.75$). However, altering the more distant variable $V_1$ guarantees the outcome ($P(Y=1 \mid V_1 \coloneq 1) = 1.0$).
>
> This demonstrates that the implications of distance are complex and non-obvious; the true influence depends on the functional interactions defined by the structural equations, not merely on the variable's position in the sequence.
>
> > are interventions consecutive or disjoint in general?
>
> In general, optimal interventions are disjoint, while they can be consecutive depending on the specific causal structure and decision requirements.

---

> ### Author Response · Authors · 2025-11-26
> **Response to Reviewer NVBs (Part 4)**
>
> > frequency of achieving S in your simulations?
>
> We report the comparison of the frequency of successfully achieving the desired domain (i.e., success rate) against the baselines (Observe, MIS, VoC) below:
>
> | Task   | Observe       | MIS           | VoC           | Ours          |
> | ------ | ------------- | ------------- | ------------- | ------------- |
> | Trader | 0.3834±0.0369 | 0.5055±0.0736 | 0.5320±0.0727 | 0.6211±0.0905 |
> | Farmer | 0.1103±0.0466 | 0.5670±0.1359 | 0.5717±0.1386 | 0.5794±0.1254 |
> | Doctor | 0.3947±0.0487 | 0.5131±0.0641 | 0.5372±0.0405 | 0.6569±0.0806 |
>
> The results demonstrate that our approach achieves the highest success rate in most cases. In the *Farmer* task, MIS, VoC, and our approach perform comparably. This is because the target variable in this specific task is influenced by a single critical variable, which all three methods correctly identified. These experimental results have been included in the revised manuscript.
>
>
> > non-boolean settings
>
> We also included a real-world application involving non-binary variables (*Bermuda*). The results are as follows:
>
> | Task    | Observe       | MIS           | VoC           | Ours          |
> | ------- | ------------- | ------------- | ------------- | ------------- |
> | Bermuda | 0.0236±0.0050 | 0.7506±0.0167 | 0.6344±0.0037 | 0.7845±0.0056 |
>
> The results demonstrate that our approach yields the highest frequency of achieving the desired domain in non-boolean settings as well.
>
> > size of alterations vs frequency of S?
>
> We analyzed the performance of our approach under different alteration budgets (i.e., the maximum number of variables allowed to be altered). The results are reported below:
>
> | Task   | 0             | 1             | 2             | 3             |
> | ------ | ------------- | ------------- | ------------- | ------------- |
> | Trader | 0.3723±0.0443 | 0.4990±0.0529 | 0.5222±0.0431 | 0.6211±0.0905 |
> | Farmer | 0.1043±0.0474 | 0.5714±0.1370 | 0.5727±0.1349 | 0.5794±0.1254 |
> | Doctor | 0.3936±0.0531 | 0.5089±0.0549 | 0.5234±0.0455 | 0.6569±0.0806 |
>
> The results show that performance generally improves as the number of allowed alterations increases. In the Farmer task, performance improves significantly when the alteration size increases from 0 to 1, but remains stable as the budget increases from 1 to 3. This is because the influence on the target variable in this specific task is dominated by a single critical variable; once that variable is altered, further alterations yield marginal gains.
>
> > 200 notation for \tau is really buried in the text here.
> >
> > 274 small note, but it should read "considering altering", not "considering to alter".
> >
> > 278 "despite it not being a causal ancestor"
> >
> > 420 "repeat experiment with ten times"
> >
> > Figure 2 is very small and quite difficult to read. Please replot or make larger for camera ready.
>
> We thank the reviewer for the detailed suggestions. We have incorporated all suggested corrections into the revised manuscript.

---

### Official Review · Reviewer_KrbU · 2025-10-31

**Soundness:** 1
**Presentation:** 1
**Contribution:** 2
**Rating:** 2
**Confidence:** 4

**Summary:**

The paper introduces *Influence Power*, a metric aimed at quantifying how altering a variable affects the probability that an outcome variable falls within a predefined desired region ($P(Y\in S)$).

The authors build on a Bellman-style recursive formulation and define Influence Power as the difference between the optimal success probability under intervention and the expected success under natural observation. They then propose a Monte-Carlo tree search (MCTS) approach to approximate influence power using observational data.

Experiments on small synthetic SCMs suggest that the proposed MCTS approach outperforms trivial baselines in increasing $P(Y\in S)$.

**Strengths:**

- Influence power is a novel metric that combines ideas of potential intervention effect and value-of-information analysis.
- The recursive Bellman-inspired formulation is a creative way to connect causal reasoning with sequential decision making. If done right, this could be impactful.
- Examples 1-4 clearly illustrate the core intuitions (but their extended form could be moved to the Appendix).

**Weaknesses:**

- Conceptual confusion: counterfactual vs. interventional reasoning

The AUF problem is framed as counterfactual ("given that the world will look like X, what if we changed Z?"), but the paper only computes interventional and observational probabilities. This gap violates the Causal Hierarchy Theorem [1]: interventional data alone cannot answer counterfactual queries. For example, in lines 11-12: "When a model predicts an undesired outcome, it is often crucial to determine what we can change to avoid it.". Posed this way, it is theoretically impossible to solve the AUF problem (at least exactly). The formulation in the paper, though, is closer to a sequential decision process (or a planning problem) than to counterfactual inference. The authors likely need to just write their problem more formally to make this clear. See more points below around assumptions and the concept of time.

[1] Bareinboim et al. "On Pearl's Hierarchy and the Foundations of Causal Inference", 2020


- Ambiguous use of time and ordering

The paper implicitly assumes a temporal sequence of variables $(V_1,\dots,V_d,Y)$ but **SCMs are static**.
Phrases such as "subsequent variables" or "before Y is finalized" suggest temporal evolution, yet this is never formalized (no explicit time, transition, or policy definition).
The authors effectively simulate time by imposing a topological order, conflating causal order with temporal decision order (see below for discussion of ACE suitability to this setting). To resolve this ambiguity, the authors need to formalize their problem within a framework that supports time (Example unclear question: Are all variables observed before Y is realized? I would guess yes, but it's very unclear).


- Unclear problem definition and assumptions

Key assumptions are not clearly stated:

1. Causal sufficiency (no hidden confounders) seems assumed but contradicted by examples involving an unobserved $U$. For instance, in example 4 the agent must choose $X$ without access to $U$, even though $U$ influences both $X$ and $Y$. From the decision-maker's perspective, $U$ acts as a hidden confounder, creating an inconsistency with the assumption of causal sufficiency (the time dimension is what makes this so complicated).
2. Is the causal order taken as known? (I assume yes after reading Proposition 1)
3. Positivity/overlap conditions for feasible alterations are unstated.
4. The predictor $h(x)$ is mentioned in preliminaries but never used formally.

The absence of assumptions makes the formal objective of the AUF problem ambiguous.


- Misinterpretation of causal relationships

Several claims conflict with standard causal semantics. Some examples:

1. *Example 2:* the paper concludes that altering $X$ is "counterproductive" despite a positive ACE. In standard SCMs, this is impossible. The claim arises only because intervening on $X$ destroys information useful for later decisions (N.B. SCMs don't support time and order of decisions).
2. *Example 4:* a non-ancestor variable $W$ is said to have positive influence on $Y$. In causal terms, $do(W)$ cannot affect $Y$ at all. The "influence" is because of the effect of $W$ on how informative later observations are, not on the data-generating process itself.

Hence, "influence" in this paper measures more like *policy utility* rather than *causal effect* (which is also why comparing to ACE is not very suitable, see next point).


- Over-reliance on ACE comparisons and why ACE is unsuitable for the current formulation.

The paper repeatedly contrasts Influence Power with ACE.
ACE is defined for **static SCMs**.
Comparing it to a metric that implicitly models information propagation over time is conceptually inconsistent. If the SCM were explicitly **unrolled over time** (with one variable per time step) and ACE were computed in that temporal formulation (e.g., changing $W_0$ and measuring its effect on $Y_t$), the comparison would then be meaningful, and $W_0$ would indeed be a causal ancestor of $Y_t$, even though it's not a causal ancestor of $Y_0$.

Additionally, ACE measures the *expected* change in $Y$ when a binary variable flips from 0 to 1. It is not informative when the goal is to increase the probability that $Y$ falls within an arbitrary region (even in a counterfactual setting, such reasoning is on an individual level, while ACE is population level) or when variables are non-binary. This comparison adds little insight and occasionally misleads (e.g., claiming that a variable with negligible ACE may still be "highly influential").

- Toy experiments and limited validation

1. The experiments involve only tiny binary SCMs, no non-linear or continuous settings are tested.
2. The baselines ("max-one", "max-all") are trivial (and arguably not suited for this problem without SCM unrolling), so improvement isn't that unsurprising. There should also be a comparison to a baseline that only observes.
3. No runtime analysis or sensitivity tests ($\alpha$ parameter, wrong topology, data size). For example, MCTS becomes expensive as the number of alterable variables grows, since the tree expands exponentially.
4. The claim that "a rough approximation suffices" lacks evidence.

Furthermore, the experiments evaluate success in expectation over the entire data-generating process rather than at the level of individual contexts.
In each task, the authors compute ($P(Y\in S)$) under different alteration strategies averaged across all exogenous realizations. This measures population-level effects but not the per-instance decision problem implied by the AUF formulation ("given this particular observation, what should be altered to avoid the undesired future?"). Consequently, the experiments do not validate whether influence power improves decision-making for specific predicted outcomes. They only show population-level improvements. This is misaligned with what AUF is framed as, a context-specific (conditional) decision problem.

- The paper includes no reproducibility statement or supplementary material. Although not strictly required, this limits my ability to verify the reported experimental results.

**Questions:**

- Are you solving a *counterfactual* question ("what if we changed X in this observed world?") or a *decision-theoretic* question ("which variable should we alter next to maximize expected success")?

- Are all variables observed before Y is realized?

- Clarify the formal problem and specify all assumptions clearly.

- Improve the experimental design.

If you choose to use ACE, unroll the SCM over time to make it fair. Consider also adding the following:
1. Add tasks with non-binary variables, nonlinear or multiplicative mechanisms, or denser graphs.
2. Add a baseline that only observes.
3. Include sensitivity analysis for $\alpha$, order errors, and sample size.
4. Report runtimes and sensitivity tests

- How stable is influence estimation across multiple Monte Carlo seeds or data resamples?

- Do you think that, given the exponential growth of the search tree in MCTS, the method is inherently limited to a small number of alterable variables (which is likely partly why the experiments use small toy models)? Perhaps you have future work in mind on sampling or pruning strategies to improve scalability?

- Notes on improving notation (minor):
1. The notation $(V_i^a=v_i) / (V_i^o=v_i)$ is non-standard. The paper introduces both this notation and the standard *do*-notation (lines 87-88). Ideally, introduce only one and use it consistently.
2. Keep observed context (x) explicit in all conditional expressions.

---

> ### Author Response · Authors · 2025-11-26
> **Response to Reviewer KrbU (Part 1)**
>
> Thank you for the insightful feedback. We hope the following responses can fully address your concerns; please let us know if any points remain unclear.
>
> ---
>
> > Conceptual confusion: counterfactual vs. interventional reasoning. The AUF problem is framed as counterfactual ...
>
> There might be a fundamental misunderstanding regarding our work. The AUF problem is ***not*** framed as counterfactual reasoning (i.e., *what would have happened, had we chosen differently at a point in the past* [1]). Instead, it is a *"**forward-looking**"* problem (i.e., *planning for the future*), as stated in the abstract. Thus, the Causal Hierarchy Theorem is ***not*** violated in this work, as we operate in the interventional layer rather than the counterfactual layer.
>
> To clarify further, in lines 11-12 of the initial submission, we stated: "When a predictive model anticipates an undesired *future* event, a question arises: what can we do to avoid it?" Posed in this way, it should be evident that the focus is on changing the future, not the past. Moreover, line 213 of the initial submission explicitly noted that "anything that has already occurred is immutable."
>
> To preclude potential ambiguity, we have refined the problem definition in the revised manuscript to emphasize this distinction more clearly.
>
> ---
>
> > Ambiguous use of time and ordering
>
> Thank you for this thoughtful question. We would like to clarify that we did *not* conflate temporal order with causal order. Instead, we considered a pre-specified sequence of variables $(V_1, \ldots, V_d, Y)$. Since this sequence constitutes a *total order*, it naturally accommodates a temporal interpretation.
>
> Crucially, to ensure the sequence of variables satisfies the formal definition of *Statistical Time* (see Definition 2.8.1 in Pearl (2009) [2]), we explicitly emphasized in line 102 of the initial manuscript that the variables are causally ordered (i.e., the total order agrees with the underlying causal structure). This has provided the theoretical formalization required to interpret the sequence as a temporal sequence. We initially omitted a detailed discussion on the semantics of Statistical Time to avoid an extended digression into *Physical Time* and the *Temporal Bias* conjecture (see Conjecture 2.8.2 in Pearl (2009) [2]). To prevent potential ambiguity, we have revised the manuscript to clarify this temporal interpretation within our framework.

---

> ### Author Response · Authors · 2025-11-26
> **Response to Reviewer KrbU (Part 2)**
>
> > Unclear problem definition and assumptions
>
> - On causal sufficiency.
>   - We explicitly stated the assumption of causal sufficiency in Proposition 1, but we chose not to assume it for the rest of the paper. Technically, causal sufficiency would be important for reliably estimating interventional probabilities from observational data. However, once the probabilities are estimated, some variables can remain unobserved when deciding whether to alter a variable.
>   - Causal sufficiency is required for estimation but is not assumed at the time of decision. We argue that this separation is more practical than assuming full observability for both phases. Consider a medical example with four ordered variables: genetic predisposition to allergies ($G$), skin condition ($X$), drug concentration ($Z$), and health recovery ($Y$). A doctor may have access to a complete historical observational dataset (with no missing values) to analyze the causal relationships and estimate the probabilities. However, in clinical practice, when treating a new patient, inspecting the genetic marker $G$ might be time-consuming or unaffordable, whereas observing the skin condition $X$ is immediate and cheap. When deciding whether to alter the drug concentration $Z$, the doctor may observe $X$ but leave $G$ unobserved. This flexibility allows the set of observed variables at decision time to be dictated by specific practical constraints.
> - On causal order.
>   - We assume that a sequence of ordered variables (i.e., a total order) is pre-specified to the decision-maker and is consistent with the underlying causal structure. This ensures that the formulation of influence power in our framework is well-defined.
> - On positivity condition.
>   - Thank you for pointing this out. The positivity condition is necessary for the reliable estimation of interventional probabilities.
> - On problem definition.
>   - In the AUF problem, given an initial observation of any subset of variables $\mathbf{X}$ (including the case of empty set $\mathbf{X} = \emptyset$), the decision-maker's goal is to positively influence the target variable $Y$ by altering a set of subsequent variables $\mathbf{Z}$. This indicates that the variables in $\mathbf{Z}$ must come after those in $\mathbf{X}$ in the sequence of variables. This restriction settles our work in the interventional layer (Layer 2 of the Causal Hierarchy), without addressing counterfactual queries (Layer 3).
>   - The predictor $h(\mathbf{x})$ refers to a prediction of $Y$ given $\mathbf{X} = \mathbf{x}$. This is formulated as (conditional) probabilities $P(Y | \mathbf{X})$ to align with the expression of (conditional) interventional probabilities.
>
>
> To enhance clarity, we have refined the problem definition and assumptions in the revised manuscript. Relaxing these assumptions further is a compelling avenue for future work.

---

> ### Author Response · Authors · 2025-11-26
> **Response to Reviewer KrbU (Part 3)**
>
> > Misinterpretation of causal relationships. Several claims conflict with standard causal semantics ...
>
> We would like to emphasize that our claims do *not* conflict with standard causal semantics; rather, our work builds upon and complements them. The observation that existing terms are not directly suited for articulating the "influence" in the context of the AUF problem is not a limitation of this work but rather highlights its strength, addressing a critical gap in the current literature.
>
> This work adopts the concept of the SCM to describe how *nature* assigns values to variables of interest, i.e., the physical mechanisms governing the natural generation process of random variables. While the SCM does not inherently encode a decision order, our framework augments it with a pre-specified sequence of variables. This augmentation is precisely what enables a well-defined notion of "influence power" for AUF problems involving multiple actionable variables.
>
> - Example 2
>   - We respectfully disagree with the reviewer's claim that "in standard SCMs, this (altering X is counterproductive despite a positive ACE) is impossible." This phenomenon can indeed occur in a standard SCM. Consider the following example: $U \sim \text{Bern}(0.1)$, $X:=1-U$, $Y:=X+U-2XU$, where $U$ is unobserved, $X$ is actionable with feasible domain $\Delta_X = \{0,1\}$, and $Y$ is the target variable (with the goal of maximizing the probability of $Y=1$, i.e., the expectation of $Y$).
>     - *Natural State*: In the observational setting, where $X$ always takes the opposite value of $U$, $Y$ is consistently equal to $1$. Thus, $\mathbb{E}(Y)=1$.
>     - *Average Causal Effect*: The ACE is $\tau(X,Y)=\mathbb{E}(Y|do(X=1))-\mathbb{E}(Y|do(X=0))=0.9-0.1=0.8$, which is positive. However, any alteration of $X$ is counterproductive, as $\mathbb{E}(Y|do(X=0)) < \mathbb{E}(Y|do(X=1)) < \mathbb{E}(Y)$.
>     - *Influence Power*: In contrast, our proposed quantity captures this: $\dot{p}(X,Y)=\max_{x}\mathbb{E}(Y|do(X=x)) - \mathbb{E}(Y) = -0.1$. This negative value properly reflects that variable $X$ should not be altered. It is worth noting that, in this simple binary case with a single actionable variable, the formulation of $\tau(X,Y)$ is close to that of $\dot{p}(X,Y)$.
> - Example 4
>   - We agree with the reviewer that the "influence" in the AUF context is not solely attributable to the physical data-generating process. As highlighted in the manuscript (lines 58-60), the proposed quantity is explicitly designed to account for the impact of alteration throughout the decision process. Therefore, this quantity is not the "causal influence" in a narrow, mechanistic sense; rather, it encompasses a broader scope that incorporates both the underlying SCM and decision dynamics, reflecting how current decisions shape the future possibilities of downstream variables constrained by the underlying SCM.
>
>
> ---
>
> > Over-reliance on ACE comparisons and why ACE is unsuitable for the current formulation
>
> We chose to compare with ACE because: 1) it is one of the most canonical and representative measures of causal strength in the literature, and 2) its formulation closely resembles the proposed quantity for AUF, as highlighted in the clarification of Example 2 above.
>
> We agree with the reviewer that ACE is defined for static SCMs, measuring causal strength through atomic interventions under the assumption that all subsequent variables evolve according to their natural mechanisms. In realistic scenarios, static causal structures often struggle to capture the dynamics of the decision process. While explicitly "unrolling" the SCM over time could adapt ACE to address the AUF problem, such unrolling itself is non-trivial and, to the best of our knowledge, no direct method currently exists to do so for the AUF problem (a related area of study is causal time series, where all the variables appear at *every* time step, which is quite different from our setting). Actually, our proposed notion of influence power can be seen as implicitly performing this intricate unrolling in a principled manner.
>
> Additionally, we would like to point out that we have taken steps to ensure the ACE is mathematically comparable to the proposed quantity. Specifically, in Section 3.3, we restricted our focus to binary variables with the target region $\mathcal{S}=\{1\}$, making the maximization of the probability of $Y \in \mathcal{S}$ equivalent to maximizing the expectation of $Y$. Besides, even when $Y$ is non-binary and $\mathcal{S}$ is arbitrary, we can establish comparability by defining an indicator variable $Y' := \mathbb{I}(Y \in \mathcal{S})$. Since $P(Y \in \mathcal{S}) = \mathbb{E}[Y']$, the ACE on $Y'$ becomes more comparable to the proposed quantity. Since defining a unified metric for causal strength in general, non-linear settings remains an open problem, we leave further exploration and comparison for future work.

---

> ### Author Response · Authors · 2025-11-26
> **Response to Reviewer KrbU (Part 4)**
>
> > Are you solving a counterfactual question ("what if we changed X in this observed world?") or a decision-theoretic question ("which variable should we alter next to maximize expected success")?
>
> This work does not address counterfactual queries (reasoning about alternative pasts for a specific instance). Instead, it focuses on decision-making in the AUF problem, determining which variable to alter next to maximize the probability of success.
>
> ---
>
> > Are all variables observed before Y is realized?
>
> As we have clarified, causal sufficiency is required for estimation but is not assumed at the time of decision.  In the historical observational data (used for estimation), all variables, including $(V_1, \ldots, V_d)$ and $Y$, are fully observed. However, during decision-making (when deciding whether to alter a variable), some variables may remain unobserved before $Y$ is realized. For example, in the Doctor task presented in our experiments, the variable $U$ is not observed during decision-making.
>
> ---
>
> > Clarify the formal problem and specify all assumptions clearly.
>
> Thank you for the suggestion. We have rigorously refined the problem definition and explicitly stated all technical assumptions in the revised manuscript to ensure formal clarity.

---

> ### Author Response · Authors · 2025-11-26
> **Response to Reviewer KrbU (Part 5)**
>
> > Improve the experimental design
>
> Thanks for the constructive suggestions. We have introduced several new baselines (Observe, MIS, VoC) and added a real-world application to strengthen our experimental evaluation. These new results have been included in the revised manuscript.
>
> The performance comparison (success rate) for the three synthetic tasks are reported below:
>
> | Task   | Observe       | MIS           | VoC           | Ours          |
> | ------ | ------------- | ------------- | ------------- | ------------- |
> | Trader | 0.3834±0.0369 | 0.5055±0.0736 | 0.5320±0.0727 | 0.6211±0.0905 |
> | Farmer | 0.1103±0.0466 | 0.5670±0.1359 | 0.5717±0.1386 | 0.5794±0.1254 |
> | Doctor | 0.3947±0.0487 | 0.5131±0.0641 | 0.5372±0.0405 | 0.6569±0.0806 |
>
> Here, *Observe* denotes a baseline that only observes without alterations. *MIS* alters a variable if it belongs to the minimal intervention set proposed by Lee and Barenboim (2018). *VoC* alters a variable when doing so increases the AUF probability of altering the subsequent variable.
>
> The results demonstrate that our approach achieves the best performance in most cases. In the *Farmer* task, MIS, VoC, and our approach perform comparably because the target variable in this specific task is influenced by a single critical variable, which all three methods correctly determined.
>
> The average runtime per single execution (in seconds) is reported below.
>
> | Task   | Observe | MIS    | VoC    | Ours   |
> | ------ | ------- | ------ | ------ | ------ |
> | Trader | 0.2968  | 1.0192 | 2.0904 | 5.3361 |
> | Farmer | 0.2124  | 0.8945 | 1.8140 | 3.3566 |
> | Doctor | 0.2679  | 0.9206 | 2.0073 | 5.2135 |
>
> As suggested, we have analyzed the performance of our approach with varying sample sizes (i.e., the number of observational instances used for estimation) as follows.
>
> | Task   | 10            | 100           | 500           | 1000          | 10000         |
> | ------ | ------------- | ------------- | ------------- | ------------- | ------------- |
> | Trader | 0.4342±0.0597 | 0.4927±0.0959 | 0.5948±0.1070 | 0.6211±0.0905 | 0.6235±0.0929 |
> | Farmer | 0.2021±0.1029 | 0.5332±0.1424 | 0.5654±0.1405 | 0.5794±0.1254 | 0.5759±0.1282 |
> | Doctor | 0.4386±0.0478 | 0.4858±0.0729 | 0.6448±0.0960 | 0.6559±0.0806 | 0.6546±0.0894 |
>
> The performance generally improves as sample size increases, stabilizing around 1,000 samples. We also evaluated the sensitivity of our MCTS method to the exploration parameter $\alpha$:
>
> | Task   | 0.01          | 0.1           | 0.5           | 1.0           | 2.0           | 5.0           | 10.0          |
> | ------ | ------------- | ------------- | ------------- | ------------- | ------------- | ------------- | ------------- |
> | Trader | 0.5584±0.0769 | 0.5808±0.0745 | 0.6015±0.0437 | 0.6104±0.0548 | 0.6273±0.0540 | 0.5915±0.0369 | 0.5904±0.0392 |
> | Farmer | 0.5579±0.1146 | 0.5595±0.1466 | 0.5793±0.1346 | 0.5736±0.1275 | 0.5622±0.1336 | 0.5655±0.1228 | 0.5550±0.1386 |
> | Doctor | 0.5831±0.0593 | 0.5990±0.0568 | 0.6675±0.0714 | 0.6592±0.0906 | 0.6560±0.0611 | 0.6431±0.0812 | 0.5822±0.0536 |
>
> The results indicate robust performance for $\alpha$ values between 0.5 and 5.0. Extreme values (too small, e.g., 0.01, or too large, e.g., 10.0) degrade performance.
>
> In addition, we have included *Bermuda*, a real-world application with non-binary variables. The performance comparison (success rate) is as follows.
>
> | Task    | Observe       | Max-One       | Max-All       | MIS           | VoC           | Ours          |
> | ------- | ------------- | ------------- | ------------- | ------------- | ------------- | ------------- |
> | Bermuda | 0.0236±0.0050 | 0.6122±0.0091 | 0.7268±0.0260 | 0.7506±0.0167 | 0.6344±0.0037 | 0.7845±0.0056 |
>
> These results further demonstrate the superiority of our approach beyond binary settings.
>
> > How stable is influence estimation across multiple Monte Carlo seeds or data resamples?
>
> We evaluated the stability of the estimated influence power under two conditions: varying only Monte Carlo seeds (fixed data), and varying both Monte Carlo seeds and data resamples. The estimation results are as follows:
>
>
> | Task   | Across multiple Monte Carlo seeds | Across multiple Monte Carlo seeds & data resamples |
> | ------ | --------------------------------- | -------------------------------------------------- |
> | Trader | -0.1446±0.0243                    | -0.1453±0.0353                                     |
> | Farmer | 0.0090±0.0124                     | 0.0084±0.0183                                      |
> | Doctor | 0.1442±0.0213                     | 0.1498±0.0260                                      |
>
> As expected, the standard deviation is slightly larger when data resampling is involved. Overall, the estimation of influence power remains stable across both conditions.

---

> ### Author Response · Authors · 2025-11-26
> **Response to Reviewer KrbU (Part 6)**
>
> >  Do you think that, given the exponential growth of the search tree in MCTS, the method is inherently limited to a small number of alterable variables (which is likely partly why the experiments use small toy models)? Perhaps you have future work in mind on sampling or pruning strategies to improve scalability?
>
> Thank you for this thoughtful question. We adopted the Monte-Carlo simulation precisely as a practical solution to approximate the influence power. Regarding scalability to larger and more complex real-world AUF problems, we agree that this is a vital direction. We can draw inspiration from the success of MCTS in domains like Go (e.g., AlphaGo), where neural networks are used as heuristics to guide the search and prune the tree [3]. Additionally, parallelization (i.e., running multiple simulations simultaneously) can significantly improve efficiency without sacrificing performance [4]. In this submission, our primary contributions focus on the problem formulation, conceptual definitions, and prototype validation. Extending the approach to handle large-scale applications is an interesting and promising direction that we leave for future work.
>
> ---
>
> > The notation $(V_i^a = v_i) / (V_i^o = v_i)$ is non-standard. The paper introduces both this notation and the standard *do*-notation (lines 87-88). Ideally, introduce only one and use it consistently.
>
> We originally employed the notation $V_i \overset{\textit{a}}{=} v_i$ to specifically denote that $V_i$ *can* be altered to $v_i \in \Delta_{V_i}$ by the decision-maker. This notation is only used when such an alteration is feasible (i.e., the alteration is allowed within the feasible domain and the variable has not yet occurred), while the standard *do*-notation allows specifying interventions regardless of whether the alteration is feasible or not.
>
> We understand the importance of standardizing notation to prevent confusion. Therefore, we have updated the manuscript: we replaced $V_i \overset{\textit{a}}{=} v_i$ and $V_i \overset{\textit{o}}{=} v_i$ with $V_i \coloneq v_i$ (for alteration) and $V_i = v_i$ (for observation), respectively.
>
> ---
>
> > Keep observed context (x) explicit in all conditional expressions.
>
> Thank you for the suggestion. We have explicitly included the context $\mathbf{x}$ in relevant conditional expressions in the revised manuscript.
>
> ---
>
> **Reference**
>
> [1] Pearl, Judea, Madelyn Glymour, and Nicholas P. Jewell. Causal inference in statistics: A primer. John Wiley & Sons, 2016.
> [2] Pearl, Judea. Causality. Cambridge university press, 2009.
> [3] Silver, David, et al. "Mastering the game of go without human knowledge." nature, 550.7676 (2017): 354-359.
> [4] Fern, Alan, and Paul Lewis. "Ensemble monte-carlo planning: An empirical study." Proceedings of the International Conference on Automated Planning and Scheduling. Vol. 21. 2011.

---

### Official Review · Reviewer_eoMz · 2025-11-01

**Soundness:** 3
**Presentation:** 3
**Contribution:** 2
**Rating:** 4
**Confidence:** 3

**Summary:**

This paper proposes a framework for the “avoiding undesired future” (AUF) problem, which aims to identify which variables should be changed to prevent a predicted negative outcome. The authors introduce a metric, called influence power, to measure how useful each variable is to achieving this goal. They claim that influence power differs from traditional causal measures such as the average causal effect,  since variables with strong causal effects may have little or even a negative influence. In contrast, weakly causal or non-causal variables may still have a positive impact. The paper also introduces a Monte Carlo Tree Search (MCTS) method to estimate influence power using observational data.

**Strengths:**

The paper tackles a relevant and important conceptual problem: moving from passive prediction to proactive intervention to avoid undesired outcomes. The motivation is to develop a principled approach to determine which variables to alter. I liked the connection between causal reasoning and utility-based decision theory, specifically through the principle of maximum expected utility and a Bellman-style recursive definition.

**Weaknesses:**

* The notion of "influence power" should be compared and contrasted with similar intervention-based approaches: for example, actual causes [1] (smallest set of variables that can be altered to change an outcome), counterfactual explanations [2] (set of interventions that optimise a counterfactual outcome), or the paper [3] on agent incentives (which also uses utility to evaluate interventions, and discusses a related notion of value of control).

* The experimental section is limited to three toy models (trader, farmer, and doctor). The baseline comparisons focus on simple strategies (altering the highest probability variables or altering all variables); it'd be good to compare against a selection of the above-recommended approaches.


[1] Halpern, Joseph Y. "A modification of the Halpern-Pearl definition of causality." arXiv preprint arXiv:1505.00162 (2015).

[2] Tsirtsis, Stratis, Abir De, and Manuel Rodriguez. "Counterfactual explanations in sequential decision making under uncertainty." Advances in Neural Information Processing Systems 34 (2021): 30127-30139.

[3] Everitt, Tom, Ryan Carey, Eric D. Langlois, Pedro A. Ortega, and Shane Legg. "Agent incentives: A causal perspective." In Proceedings of the AAAI conference on artificial intelligence, vol. 35, no. 13, pp. 11487-11495. 2021.

**Questions:**

* Can the authors clarify how their "influence power" notion and overall framework relate to the above-suggested approaches?

* Given the computational intensity and large number of Monte Carlo simulations required to achieve meaningful results, can the authors clarify how their method scales in practice?

---

> ### Author Response · Authors · 2025-11-26
> **Response to Reviewer eoMz (Part 1)**
>
> Thank you for the constructive feedback. We hope the following responses can fully address your concerns; please let us know if any points remain unclear.
>
> ---
>
> > Can the authors clarify how their "influence power" notion and overall framework relate to the above-suggested approaches?
>
> There are substantial differences between our work and the aforementioned approaches. The most notable distinction is that [1], [2], and [3] deal with **counterfactuals** (i.e., *reasoning about the past*), whereas the AUF problem addressed in our work is **forward-looking** (i.e., *planning for the future*). Specifically:
>
> - Contrast with [1]: Reference [1] focuses on reasoning about a speciﬁc cause $X$ of an observed outcome $Y$ by asking what $Y$ would have been if $X$ had been different. In contrast, our work aims to influence a future target $Y$ by altering an actionable variable $Z$, neither of which has occurred yet. In the AUF (Avoiding Undesired Future) problem, altering variables that have already occurred is not allowed, as the past cannot be changed in reality.
> - Contrast with [2]: The work of [2] is related to our work due to its connection with the Bellman equation, but it fundamentally differs from our work in three key aspects: 1) *Objective*: Like [1], [2] focuses on counterfactual reasoning (i.e., asking what would have happened in the past if actions had been different), whereas our work focuses on forward-looking planning (i.e.,g determining what to do for influencing the future). 2) *Variable definition*: [2] maintains a strict dichotomy between state variables and action variables. In contrast, our framework treats all variables uniformly as random variables, allowing them to be flexibly observed or altered without such a rigid distinction. 3) *Model requirements*: Computing counterfactual probabilities in [2] requires knowing the underlying structural causal model (SCM). In contrast, our approach estimates interventional probabilities without assuming that the SCM is known.
> - Contrast with [3]: Reference [3] primarily analyzes an existing concept called Value of Control (VoC), proposed by Schachter (1986), and introduces a new concept, Instrumental Control Incentive (ICI), which is defined on nested counterfactuals. [3] differs significantly from our work for the following reasons: 1) In [3], a decision node has no "natural value" by definition; its value is entirely assigned by the decision-maker. In contrast, the alteration operation in our framework is applied to random variables, whose natural generation processes are governed by the underlying SCM; one can choose to intervene (set the value) or refrain from intervening (letting the variable occur naturally). 2) [3] is restricted to a single decision node, while our framework accommodates an arbitrary number of actionable variables. *These differences are significant, leading to novel conclusions that are not discovered before.* For instance, [3] only shows that variables with a directed path to the target variable can have a positive value of control, while we find that a variable that is not an intrinsic ancestor of the target variable in the underlying SCM can have a positive influence power.
>
> We have incorporated these comparisons into the related work section of the revised manuscript.

---

> ### Author Response · Authors · 2025-11-26
> **Response to Reviewer eoMz (Part 2)**
>
> > The experimental section is limited to three toy models (trader, farmer, and doctor). The baseline comparisons focus on simple strategies (altering the highest probability variables or altering all variables); it'd be good to compare against a selection of the above-recommended approaches.
>
> Thanks for the suggestions. In response, we have introduced several new baselines and included a real-world application (*Bermuda*) from the literature to strengthen our experimental evaluation. Due to the fundamental differences in objectives and formalizations, the approaches in [1], [2], and [3] cannot be directly applied to the AUF problem. Nevertheless, to facilitate a meaningful comparison, we have adapted the concept of VoC analyzed in [3] into our framework.
>
> The performance (success rate) on the four tasks are presented below:
>
> | Task   | Observe       | MIS           | VoC           | Ours          |
> | ------ | ------------- | ------------- | ------------- | ------------- |
> | Trader | 0.3834±0.0369 | 0.5055±0.0736 | 0.5320±0.0727 | 0.6211±0.0905 |
> | Farmer | 0.1103±0.0466 | 0.5670±0.1359 | 0.5717±0.1386 | 0.5794±0.1254 |
> | Doctor | 0.3947±0.0487 | 0.5131±0.0641 | 0.5372±0.0405 | 0.6569±0.0806 |
> | Bermuda | 0.0236±0.0050 | 0.7506±0.0167 | 0.6344±0.0037 | 0.7845±0.0056 |
>
> Here, *Observe* denotes a baseline that only observes, as suggested by Reviewer KrbU. *MIS* alters a variable if it belongs to the minimal intervention set proposed by Lee and Barenboim (2018). *VoC* alters a variable when doing so increases the AUF probability of altering the next variable.
>
> The results demonstrate that our approach achieves the best performance in most cases. In the *Farmer* task, MIS, VoC, and our method perform comparably. This is because the target variable in this specific task is influenced by a single critical variable, which all three methods correctly determined. These new experimental results have been included in the revised manuscript.
>
> ---
>
>
> > Given the computational intensity and large number of Monte Carlo simulations required to achieve meaningful results, can the authors clarify how their method scales in practice?
>
> Thank you for this thoughtful question. In this work, we employed a classic Monte-Carlo Tree Search (MCTS) algorithm to efficiently approximate the proposed quantity, as the complexity of exact computation grows exponentially with the number of variables. To address scalability in practice, we can leverage well-established techniques from the MCTS literature to scale our approach to larger, complex real-world AUF problems. For example, parallelization (i.e., running multiple simulations simultaneously) can significantly improve efficiency without sacrificing performance [4]. Additionally, drawing on the success of MCTS in high-complexity domains such as board games (e.g., AlphaGo [5]), neural networks can be integrated to guide simulations, effectively pruning the search space and further enhancing efficiency.
>
> In this submission, our primary contributions focus on the problem formulation, conceptual definitions, and prototype validation. Extending the approach to larger-scale problems is an interesting and promising avenue for future research.
>
> ---
>
> **Reference**
>
> [1] Halpern, Joseph Y. "A modification of the Halpern-Pearl definition of causality." arXiv preprint arXiv:1505.00162 (2015).
> [2] Tsirtsis, Stratis, Abir De, and Manuel Rodriguez. "Counterfactual explanations in sequential decision making under uncertainty." Advances in Neural Information Processing Systems 34 (2021): 30127-30139.
> [3] Everitt, Tom, Ryan Carey, Eric D. Langlois, Pedro A. Ortega, and Shane Legg. "Agent incentives: A causal perspective." In Proceedings of the AAAI conference on artificial intelligence, vol. 35, no. 13, pp. 11487-11495. 2021.
> [4] Fern, Alan, and Paul Lewis. "Ensemble monte-carlo planning: An empirical study." Proceedings of the International Conference on Automated Planning and Scheduling. Vol. 21. 2011.
> [5] Silver, David, et al. "Mastering the game of go without human knowledge." nature, 550.7676 (2017): 354-359.

---

> > ### Comment · Reviewer_eoMz · 2025-11-27
> >
> > Thank you for addressing my points. I have two follow-ups:
> >
> > 1) Since you intervene only in the future (i.e., no counterfactuals), then your approach can be applied beyond SCMs, i.e., to Causal Bayesian networks where each equation $Y = f_Y(pa_Y,N_Y)$ is replaced by a conditional distribution $Y \sim P(Y \mid pa_Y)$. (since you don't need counterfactuals, you don't need to isolate the noise variables) This would make your approach more generally applicable, no?
> >
> > 2) I'd like to understand more about the Bermuda case study.
> > * I've seen the description in A.2, but it'd be good to write down in that section also the actual structural equations and a causal graph.
> > *  I understand you built this model (i.e., it wasn't pre-existing), so it'd be good to provide some more details about the model design. For instance, were causal relationships already known or have you discovered them from data?
> > * Are the five variables you can manipulate actionable in real life, or have you chosen them just for demo purposes?

---

> ### Author Response · Authors · 2025-11-27
> **Response to Follow-up Comments**
>
> Thank you for your prompt reply and follow-up questions. We address your specific points below.
>
> ---
>
> > Since you intervene only in the future (i.e., no counterfactuals), then your approach can be applied beyond SCMs, i.e., to Causal Bayesian networks where each equation $Y=f_Y(pa_Y,N_Y)$ is replaced by a conditional distribution $Y \sim P(Y|pa_Y)$. (since you don't need counterfactuals, you don't need to isolate the noise variables) This would make your approach more generally applicable, no?
>
> Thank you for this insightful observation. The reviewer is correct that our approach is general and applies to Causal Bayesian Networks (CBNs) as well as Structural Causal Models (SCMs). We focused our formalization on SCMs because:
>
> 1. **Modeling Physical Mechanisms:** SCMs are conceptually distinct in their focus on "modeling physical mechanisms" [6] and emphasizing "how *nature* assigns values to variables" [7]. This ontological perspective is crucial for the central insight of our paper: a variable that is not an ancestor of the target variable in the *natural* generation process may still possess positive influence power for avoiding the undesired future.
> 2. **Intuition and Determinism:** SCMs are often considered "more in tune with human intuition" than CBNs [8]. We note that Pearl originally advocated for SCMs to describe causal relationships because they reflect Laplace’s conception of natural phenomena, making them more suitable for modeling physical mechanisms in nature (see the beginning of Section 1.4 in Pearl (2009) [8]).
>
> While SCMs are versatile and useful for tasks such as counterfactual reasoning and decision-process modeling, our motivation for adopting SCMs aligns specifically with Pearl's perspective on modeling natural phenomena.
>
> ---
>
> > I've seen the description in A.2, but it'd be good to write down in that section also the actual structural equations and a causal graph.
>
> Thank you for this suggestion. We have now included a description of the causal graph and the actual structural equations for the *Bermuda* case study in the revised manuscript.
>
> > I understand you built this model (i.e., it wasn't pre-existing), so it'd be good to provide some more details about the model design. For instance, were causal relationships already known or have you discovered them from data?
>
> The underlying SCM for the *Bermuda* case study was adopted from existing causal literature [9]. Specifically, the causal relationships were already provided by previous ecological studies [10], and the structural equations were then obtained by performing linear regression on the 50 observations provided by [11]. We have clarified this origin in the revised text.
>
> > Are the five variables you can manipulate actionable in real life, or have you chosen them just for demo purposes?
>
> The five actionable variables were selected to align with the setup in the existing literature [9] and were chosen primarily for demonstration purposes within this benchmark.
>
> ---
>
>
>
> We hope this response addresses your concerns, and we are happy to provide further clarification or address follow-up questions.
>
> We sincerely thank you again for your time and suggestions, and for careful consideration of our response and the revised manuscript.
>
>
>
> ---
>
> **Reference (continued)**
>
> [6] Peters, Jonas, Dominik Janzing, and Bernhard Schölkopf. Elements of causal inference: foundations and learning algorithms. The MIT press, 2017.
> [7] Pearl, Judea, Madelyn Glymour, and Nicholas P. Jewell. Causal inference in statistics: A primer. John Wiley & Sons, 2016.
> [8] Pearl, Judea. Causality. Cambridge university press, 2009.
> [9] Aglietti, Virginia, et al. "Causal bayesian optimization." International Conference on Artificial Intelligence and Statistics. PMLR, 2020.
> [10] Courtney, Travis A., et al. "Environmental controls on modern scleractinian coral and reef-scale calcification." Science advances 3.11 (2017): e1701356.
> [11] Andersson, Andreas, and Nicholas Bates. "In situ measurements used for coral and reef-scale calcification structural equation modeling including environmental and chemical measurements, and coral calcification rates in bermuda from 2010 to 2012 (BEACON project)", 2018.

---

### Author Response · Authors · 2025-12-04
**Summary of Changes in Revision**

We sincerely thank the reviewers for their valuable feedback, which has significantly improved our work. We have addressed all concerns through additional experiments, clarifications, and refinements to the paper. Below is a summary of the key changes in the revised manuscript:

- [Section 2] We refined the problem definition to emphasize that the AUF problem is forward-looking (planning for the future) rather than counterfactual (reasoning about the past), and our formalization inherently accommodates a temporal interpretation.
- [Section 3.3] We visualized the relationships between the proposed and canonical concepts in Figure 2, and formally summarized these claims in Theorem 1; the full proof was deferred to Appendix B.
- [Section 4.2] We added a remark following Proposition 1 to clarify that causal sufficiency is required only for Proposition 1 and is not assumed for the rest of the paper.
- [Section 5] We added ablation studies, multiple baselines, and a real-world application from existing literature to strengthen our experimental evaluation.
- [Section 6] We included a discussion comparing our work with related approaches based on counterfactuals.

We believe these updates fully address the reviewers' concerns and significantly improve the quality and clarity of the paper. Accordingly, we trust that the revised manuscript meets the reviewers' expectations.

---

### Author Response · Authors · 2025-12-04
**Summary of Rebuttal**

We thank the reviewers for their constructive feedback.

We are encouraged by their recognition of the **novelty of our measure** (*KrbU*, *NVBs*, *rkWZ*), our contribution to **bridging causality and decision theory** (*eoMz*, *KrbU*, *rkWZ*), and the clarity of our core intuitions (*KrbU*, *rkWZ*). We specifically thank Reviewer *NVBs* for stating that they "**quite like this paper**."

Below, we outline how we addressed key concerns raised during the rebuttal:

1. **Clarification on Problem Scope** (*eoMz*, *KrbU*): We refined the problem definition to emphasize that the AUF problem is forward-looking (planning for the future) rather than counterfactual (reasoning about the past) [Section 2]. We also expanded the Related Work section to compare our work with previous counterfactual-based approaches [Section 6].
2. **Clarification on Causal Sufficiency** (*KrbU*, *rkWZ*): We added a remark following Proposition 1 [Section 4.2] emphasizing that causal sufficiency is an assumption *only* for Proposition 1, not for the general framework. This distinction ensures practicality in real-world scenarios where variables (e.g., genetic markers) may be unobserved at decision time, even if they were present in the historical training data.
3. **Expanded Experimental Validation** (*eoMz*, *KrbU*, *NVBs*): To address concerns regarding "toy models," we introduced the real-world task (Bermuda) involving non-binary variables [Section 5]. We also added three baselines (Observe, MIS, and VoC). Results demonstrate that our approach outperforms these baselines in AUF success rates.
4. **Ablation Studies** (*KrbU*, *NVBs*): We conducted sensitivity analyses on sample sizes, hyperparameters, and alteration budgets [Section 5]. Results show performance stabilizes around 1000 samples and is robust for exploration parameters $\alpha \in [0.5, 5.0]$. Additionally, we confirmed that performance scales with alteration budgets and verified estimation stability across multiple seeds and resamples.
5. **Notation and Refinements** (*NVBs*, *rkWZ*): We standardized our notation to distinguish between alteration and observation, improving overall readability.

We believe our detailed responses and additional experiments have fully addressed the concerns, and we kindly request your favorable consideration of our submission.

---

### Meta-Review · Area_Chair_EP7j · 2026-01-07

**Summary:**

Reviewers raised concerns about the positioning of the proposed notion of influence power relative to existing intervention-based methods, the clarity of the problem formulation, and the limited scope of experimental evaluation. In particular, some reviewers questioned whether the approach should be framed as counterfactual inference or sequential decision making, and noted that several experiments were conducted on toy datasets (eoMz, KrbU).

Overall, the authors addressed these concerns by clarifying the forward-looking nature of the formulation, explicitly stating the underlying assumptions, expanding the experimental evaluation with a real-world Bermuda experiment, and improving presentation and runtime analysis. Most reviewers remain positive to the paper, and the remaining concerns were not considered substantial enough to outweigh the paper’s contributions.

**Reviewer Concerns:**

### Addressed

- **Clarification of formulation and positioning:**
  The authors clarified that the proposed notion of influence power is forward-looking rather than counterfactual, addressing confusion about its relationship to existing intervention-based methods.

- **Expanded experimental evaluation:**
  Additional experiments were added, including a real-world Bermuda experiment, to move beyond purely toy settings.

- **Presentation and runtime analysis:**
  Notation issues were corrected and runtime comparisons were added, improving clarity and completeness.

### Outstanding

- **Limited empirical depth:**
  While new experiments were added, some reviewers remain concerned that the empirical evaluation is still relatively limited and may not fully demonstrate the method’s robustness across diverse, complex settings.

- **Assumptions and formulation clarity:**
  Despite added discussion, questions remain about the precise framing (e.g., counterfactual inference vs. sequential decision making) and the strength and realism of the underlying assumptions.

**Reviewer Scores:**

Reviewer **eoMz** raised concerns about how the proposed notion of influence power differs from existing intervention-based methods, and noted that the experiments were limited to three toy datasets. The authors clarified that prior approaches focus on counterfactual reasoning, whereas their method is forward-looking, and added a real-world Bermuda experiment. This response is likely to address the concern, and the reviewer may raise their score.

Reviewer **KrbU** questioned whether the formulation should be viewed as counterfactual inference or sequential decision making, and noted that key assumptions were not clearly stated and that the experiments were weak. The authors clarified the forward-looking nature of their formulation, added discussion of the assumptions, and included additional experiments. This may partially address the concerns, and the reviewer may maintain a negative score or potentially raise it.

Reviewer **NVBs** noted that the experiments are mostly toy examples but expressed an overall positive view of the paper, as acknowledged in their review. This reviewer is likely to remain positive.

Reviewer **rkWZ** raised concerns about notation clarity and requested runtime analysis. The authors addressed these issues by revising the notation and adding runtime comparisons. As a result, this reviewer is likely to remain positive.

---

### Decision · Program_Chairs · 2026-01-26

Accept (Poster)